# Medicines prescribed for asthma, discontinuation and perinatal outcomes, including breastfeeding: A population cohort analysis

**Gareth Davies**[1], **Sue Jordan**[1]*, **Daniel Thayer**[1], **David Tucker**[2], **Ioan Humphreys**[1]

**1** Faculty of Health and Life Science, Swansea University, Swansea, United Kingdom, **2** Public Health Wales, Cardiff, United Kingdom

* s.e.jordan@swansea.ac.uk

**Data Availability Statement:** All relevant data are within the manuscript and its Supporting Information files. Additional data are available in the EUROmediCAT report and its appendices

## Abstract

### Objectives

To explore associations between exposures to medicines prescribed for asthma and their discontinuation in pregnancy and preterm birth [<37 or <32 weeks], SGA [<10th and <3rd centiles], and breastfeeding at 6–8 weeks.

### Methods

**Design.** A population-based cohort study.

   **Setting.** The Secure Anonymised Information Linkage [SAIL] databank in Wales, linking maternal primary care data with infant outcomes.

   **Population.** 107,573, 105,331, and 38,725 infants born 2000–2010 with information on premature birth, SGA and breastfeeding respectively, after exclusions.

   **Exposures.** maternal prescriptions for asthma medicines or their discontinuation in pregnancy.

   **Methods.** Odds ratios for adverse pregnancy outcomes were calculated for the exposed *versus* the unexposed population, adjusted for smoking, parity, age and socio-economic status.

### Results

Prescriptions for asthma, whether continued or discontinued during pregnancy, were associated with birth at<32 weeks' gestation, SGA <10th centile, and no breastfeeding (aOR 1.33 [1.10–1.61], 1.10 [1.03–1.18], 0.93 [0.87–1.01]). Discontinuation of asthma medicines in pregnancy was associated with birth at<37 weeks' and <32 weeks' gestation (aOR 1.22 [1.06–1.41], 1.53 [1.11–2.10]). All medicines examined, except ICS and SABA prescribed alone, were associated with SGA <10th centile.

### Conclusions

Prescription of asthma medicines before or during pregnancy was associated with higher prevalence of adverse perinatal outcomes, particularly if prescriptions were discontinued

(reference 27 - publicly available). The datasets generated and analysed during the current study are not publicly available, because the anonymized data, held by SAIL, can only be accessed within the SAIL secure remote access environment within the context of an approved project, following governance review. No patient level data are available under the terms of ethical and governance reviews. No participant consent is obtained for population level studies. Individual records for all databases were anonymised, and individual patient data cannot be publicly deposited or fully shared upon request. They are only available directly from the database providers. All interested parties are able to apply to obtain the data in the same way as the current investigators. Data used in the study can be accessed within the SAIL secure environment subject to approval by the SAIL Information Governance Review Panel. The URL for the SAIL databank is: https://saildatabank.com/, and the non-author contacts are Cynthia McNerney, Information Governance Manager, The SAIL Team, Swansea University: c.l. mcnerney@swansea.ac.uk and Charlotte Arkley, Administrator, SAIL Databank c.r.arkley@swansea. ac.uk. Dataset information (names of datasets and provenance) is provided under 'Methods, subsection Setting'. As indicated in the text, further details are in reference 27. The corresponding author will endeavour to meet requests for further data and information. We confirm that data that are not publicly available are not part of the minimal data set.

**Funding:** Funding This paper was developed from the EUROmediCAT project, and uses the cohort identified in that project. The analyses presented here were completed outside the funded period. Financial support for the EUROmediCAT study was provided by the European Union under the 7th Framework Program [grant agreement HEALTH-F5-2011-260598]. Start date: 1 March 2011. Duration: 48 months. Coordinator: Prof. Helen Dolk, University of Ulster. Further information can be found at www.euromedicat.eu. The paper is based on data in the all-Wales SAIL databank, which is supported by UK Research and Innovation funding to Swansea University through an Administrative Data Research Centre grant (2018-2021), project reference: ES/S007393/1, Principal Investigator: Professor David Ford. The funder played no part in the study or production of the paper.

**Competing interests:** The authors have declared that no competing interests exist.

during pregnancy. Women discontinuing medicines during pregnancy could be identified from prescription records. The impact of targeting close monitoring and breastfeeding support warrants exploration.

## Introduction

The prevalence of asthma in pregnancy has increased worldwide [1–3]. Approximately 9% of pregnant women in the UK are prescribed medicines for asthma, more than elsewhere in Europe [4]. However, there is no consensus on the effect of asthma or prescription of asthma medicines on perinatal outcomes. While older [5] and smaller [6, 7] studies were reassuring, asthma [3, 8] accompanied by symptoms [5, 9] or suboptimal spirometry recordings [10] has been associated with growth restriction [small for gestational age [SGA]]. Premature birth is associated with asthma diagnosis in some studies [11, 12], but not others [8]. Active management of asthma may reduce premature birth [13], and uncontrolled asthma increases the risk of adverse perinatal outcomes [3], but there is less information on medicines prescribed for asthma. Few RCTs have examined the impact of asthma medicines in pregnancy on perinatal outcomes, and no differences were seen in any of the small trials located [14]. Short-Acting Beta Agonists [SABA] have not been associated with growth restriction or premature birth, although studies have relatively low numbers [15], and there is no consensus on the impact of Long-Acting Beta Agonists [LABA] [15–17]. There are few differences between medicines within each class [18]. Up to 50% pregnant women discontinue asthma medicines, often without professional advice, frequently worsening asthma and outcomes [1, 9, 14].

Medicines prescribed for asthma cross the placenta and enter breastmilk [6, 13]. Prescription of asthma medicines in early pregnancy is associated with increased prevalence of congenital anomalies [aOR 1.20, 1.08–1.34], but the effects of asthma could not be separated from those of its treatment [19]. Premature birth, SGA, and suboptimum breastfeeding threaten survival, developmental potential and ill-health from non-communicable disease in developed and developing countries [20], and the effects of medicines and their discontinuation on the range of pregnancy outcomes, including lactation, need to be considered together.

UK guidelines recommend that women prescribed asthma medicines breastfeed, and offer reassurance on oral corticosteroids [5]. Few studies reporting medicines' use and breastfeeding were located, all relatively small, and suggesting that breastfeeding rates were lower if medicines were prescribed. None considered medicines for asthma [21]. Despite the known health and economic benefits of breastfeeding [22, 23], we were unable to locate reports of the impact of prescription medicines on the prevalence of breastfeeding amongst women with asthma.

To identify women likely to benefit from additional care, and target support, we aimed to explore any associations between prescriptions for asthma medicines during pregnancy, or their discontinuation, and the range of perinatal outcomes that matter to women [prematurity, SGA, and, for the first time, breastfeeding at 6–8 weeks, full or partial].

## Methods

A population-based cohort was built from prospectively collected routine NHS data and analysed retrospectively [24]. We explored association, rather than causality, as there was no randomisation.

## Ethics

The Secure Anonymised Information Linkage [SAIL] Databank Information Governance Review Panel [IGRP] approved the study on behalf of the National Research Ethics Service, Wales on 24th March 2011. Data were irrevocably anonymised and obtained with permission of the relevant Caldicott Guardian and Data Protection Officer.

## Setting

Data were extracted from existing routinely collected data in SAIL, housed in Swansea University https://saildatabank.com/faq/ [25–27]. By 2014, ~40% of general practices had agreed to share data with SAIL, without payment. Women in the included population were slightly less deprived and older than the rest of Wales [27]. Using unique personal identifiers, which remained undisclosed to researchers to ensure anonymity, we linked primary care records, including prescriptions, to: the Office of National Statistics (ONS) births and deaths register, the National Community Child Health Database [NCCHD] [http://www.publichealthwales observatory.wales.nhs.uk/ncchd], the Patient Episode Database for Wales [http://www.wales. nhs.uk/document/176173], and CARIS [Congenital Anomaly Register and Information Service for Wales] [http://www.caris.wales.nhs.uk/home]. Databases were linked by a trusted third party [NHS Wales Informatics Service [NWIS]] [http://www.wales.nhs.uk/sitesplus/956/ home].

## Population

The study population included all births in Wales after 24 gestational weeks between 1st January 2000 and 31st December 2010, with linked maternal prescription data. Infants were included where the associated maternal ID could be linked with the primary care dataset (dependent on the general practice) and the record was complete [27]. We included all infants where the woman was present in the linked database with primary care prescription information 91 days before LMP to birth. Information on start of pregnancy was obtained from ultrasound scan data recorded in the NCCHD for Wales [27]. Infants with congenital anomalies were excluded, as associations with asthma prescriptions are reported elsewhere [19]. The impact of antidepressants on breastfeeding, premature birth and SGA in this cohort has been reported [28].

## Exposure

Prescriptions reflect physicians' assessment of asthma severity [5, 29], since prescribers adhere to guidelines [5] that direct monitoring and prescribing. Exposure to "any asthma medicines" was defined as the woman having been prescribed at least one asthma medicine [ATC code R03] in the 3 trimesters of pregnancy, defined as 1st day of LMP to birth [15]. Exposure was grouped, based on 5-digit ATC codes [30] into 10 categories, listed in Table 1. Exposure to oral corticosteroids was restricted to those co-prescribed other asthma medicines.

Combined LABA and ICS preparations [R03AK] were added to the separate exposure categories, and not analysed as one group as this would preclude discussion of the disparate biological effects. We did not analyse exposure to systemic beta2 agonists, cromolyns, theophylline or anti-cholinergic medicines separately, due to very low prescription rates in this population (<40 pregnant women exposed to any of these 2004–10) [4].

## Outcomes

**Prematurity** was defined as <37 completed weeks' gestation, and 'very premature' as <32 completed weeks [32].

**Table 1. Exposure classification.**

| Description | Medicine | ATC | BTS step |
|---|---|---|---|
| Asthma before or during pregnancy | Any asthma medicine 1 year before pregnancy or during pregnancy pre4 to t3 | R03 | Any |
| Medicated asthma | Any asthma medicines t1-t3 | R03 | Any |
| Un-medicated asthma /Prescription discontinuation | Any asthma medicine 1 year before pregnancy [pre4 to pre1], but none in pregnancy | R03 | Un-medicated asthma |
| SABA only | Inhaled Short-acting beta-2-agonists [SABA], as the only asthma medicine in pregnancy [t1-t3] | R03AC02 to R03AC04 | 1 [least severe] |
| SABA any | Inhaled Short-acting beta-2-agonists [SABA], any prescription in pregnancy [t1-t3] | R03AC02 to R03AC04 | 1 or higher |
| ICS only | Inhaled corticosteroids [ICS] as the only asthma medicine in pregnancy [t1-t3] | R03BA | 2 and well controlled |
| ICS any | Inhaled corticosteroids [ICS] any prescription in pregnancy [t1-t3] | R03BA | 2 or higher |
| LABA any | Inhaled long-acting beta-2-agonists [LABA] t1-t3 | R03AC12, R03AC13 | 3 or higher |
| LKA any | Leukotriene receptor antagonists [LKA] t1-t3 | R03DC | 3 or higher |
| OCS asthma | Oral corticosteroids [OCS] in combination with any asthma medicine [R03] t1-t3 | H02AB + [R03] | 5 [most severe] |

In the UK, asthma is initially managed with short acting beta2 agonist [SABA] bronchodilators for symptom relief [step 1]. However, if these are needed >twice per week or >once a week at night or if exacerbations have occurred in the last 2 years, inhaled corticosteroids [ICS] are recommended [step 2]. If these fail to control symptoms, *additional therapy is prescribed*, usually long acting beta2 agonists [LABA] or leukotriene antagonists [LKA], but occasionally oral beta2 agonists or theophylline [step 3]. Oral corticosteroids are prescribed if combinations of these medicines plus high dose ICS fail to control symptoms [step 5] or to manage acute exacerbations [5, 29]. British Thoracic Society [BTS]5 step 4 is characterised by high dose ICS plus step 3 medicines. We were unable to extract information on doses, so did not identify doses of oral medicines or step 4 [high dose ICS]: other inhalers are only available in standard doses. The only indications for these medicines is 'conditions associated with reversible airways obstruction' [29].

t represents trimester, t1-3 represents the time between 1st day of last menstrual period [LMP] and birth

Pre1-4 represents the 4 quarters preceding pregnancy

Pre1 represents the quarter immediately preceding pregnancy

Medicines were defined by ATC codes, and then matched to version 2 Read codes in the GP database, using reference data provided by NHS Digital Technology Reference Data Update Distribution [https://isd.digital.nhs.uk/trud3/user/guest/group/0/home] [31].

There were 0 exposures to ATCs R03AC05-7.

**Growth centiles** were calculated from WHO standards for the UK, and infants below the 10th and 3rd centiles were identified; the latter is defined as 2 standard deviations below the median or 50th centile [33].

**Breastfeeding** in this database is defined as 'any breastfeeding' (exclusive or combined, pre-dominantly or partially, with other feeds, including formula milk) as routinely recorded by health visitors at birth and 6–8 weeks and entered into NCCHD [34]. Data collection is more complete in some Health Board regions than others [35]. Breastfeeding data were available from 2004. Breastfeeding at birth may represent intention to breastfeed, rather than actuality: a high proportion of dyads breastfeeding at birth have discontinued a week later. Therefore, this measure is downgraded as evidence of successful breastfeeding [36].

## Confounding

To minimise **confounding by co-exposure**, we restricted the dataset by excluding pregnancies known to be at risk of adverse outcomes [37]. We achieved a relatively homogeneous popula-tion by excluding from the main analysis infants: 1) with major congenital anomalies; [38] 2) from multiple pregnancies; 3) stillborn; 4) exposed in the quarter preceding pregnancy or tri-mester 1 to medicines more closely associated with adverse outcomes than asthma medicines:

anti-epileptic drugs [AEDs] [NO3] [39]; coumarins [B01AA], mainly warfarin [40]; insulins [A10A]58; [41] and 5) whose mothers had any record of: heavy alcohol use and/or substance misuse [42], (many of these substances are powerful vasoconstrictors). Prevalences were checked before exclusion (tabulated in Jordan et al 2019, S3 Table).

We did not exclude moderate alcohol use as this is not known to affect perinatal outcomes [42], and may be inconsistently recorded. To minimize **confounding by indication**, we investigated both medicated and unmedicated asthma, as discontinuation of medication in pregnancy. We adjusted for socioeconomic status [SES], as Townsend fifths (S1 Table), parity, smoking [as 'yes' or 'no'], year of birth, and maternal age, grouped [43].

### Statistical analysis

We explored associations between prematurity, SGA and breastfeeding and prescription of asthma medicines 1 year before or during pregnancy, prescription discontinuation during pregnancy, and individual asthma medicines [Table 1]. For medicines with sufficient numbers of exposures, and where feasible, data were explored by logistic regression, backwards likelihood ratio [44], with covariates, SES [45], parity, smoking, age, and year of birth [S1 Table], using SPSS version 25 for windows [46]. For each analysis we compared the exposed with the unexposed population. We compared outcomes for women with medicated and un-medicated asthma (S3 Table). Odds ratios, unadjusted and adjusted are reported together with 95% confidence / compatibility intervals. We explored demographic differences between women with and without breastfeeding data at 6–8 weeks.

### Results

We identified 117,717 pregnancies with complete primary care prescription data. We excluded 4401 infants with congenital anomalies [including 582 termination of pregnancy for foetal anomalies] and 2589 multiples, leaving 110,727 singletons, of whom 390 were stillborn [41 stillborn multiples].

When those exposed to insulin, AEDs or coumarins were excluded, numbers fell to 109,299, reducing to 107,573 available for analysis when those exposed to heavy drinking or substance misuse were excluded; 105,311 and 38,725 had data on centiles and breastfeeding at 6–8 weeks [Fig 1].

Stillbirth was more prevalent amongst women prescribed asthma medicines than the unexposed population [75/13,516, 0.55% vs.431/113,316, 0.38%, OR 1.56, 1.21–2.00], particularly if medicines had been discontinued throughout pregnancy [27/3850, 0.70%, OR1.91, 1.29–2.82]. Prevalence of deprivation, smoking, substance or alcohol misuse, prescription of insulin and AEDs was higher amongst women prescribed asthma medicines, with the exception of those prescribed ICS or ICS only. The prevalence of obesity [BMI>30] was higher amongst women with asthma, particularly those prescribed corticosteroids. Women discontinuing asthma medicines were younger, but not more deprived, than those continuing [S1 Table].

Asthma medicines were prescribed to 12,690/107,573 [11.8%] of the cohort before or during pregnancy; 3589/12690 [28.3%] received their last prescription before pregnancy, and were deemed to have 'discontinued' [S2 Table]. Most [8413, 92.4%] of the 9101 women prescribed asthma medicines in pregnancy received SABA, and 3820 [45.4%] of these 8413 received only SABA. Only 3875/ 9109 (42.6%) women received ICS, and 374 received ICS only. LABA were prescribed to 465 women and LKA to 89. OCS were prescribed to 519 women co-prescribed medicines for asthma. Women with breastfeeding data were older than those without [mean ages 28.48 [6.09] vs. 28.04 [6.04] years, mean difference 0.44 [0.36–0.51], t 11.36, df 79659, p<0.001], more deprived [mean Townsend score 0.40 [3.24] vs. 0.21 [3.12], mean difference

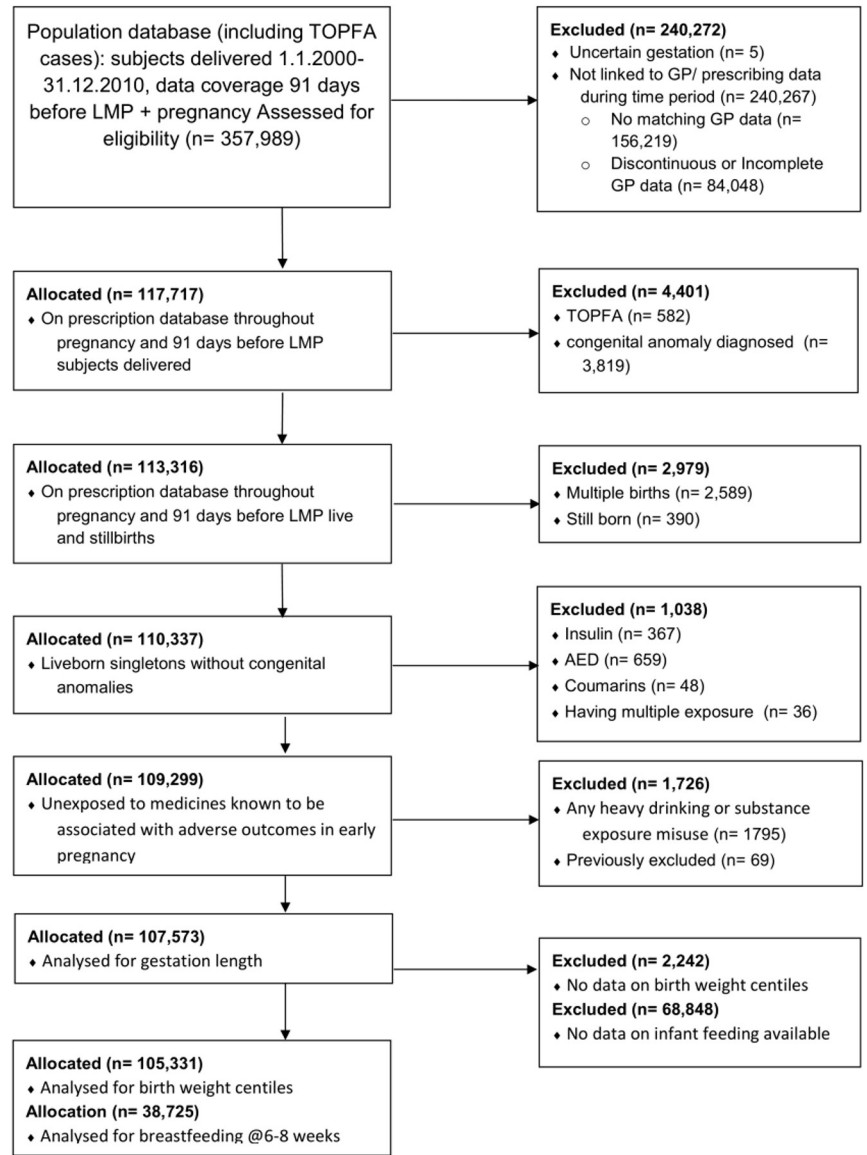

**Fig 1. Study flow diagram.**

0.20 [0.16–0.24], t 9.65, df 77578, p<0.001], and less likely to be primiparous [16095/38725, 41.6% vs. 29388/68848, 42.7% OR 0.96 [0.93–0.98].

## Premature birth

Premature birth at <37 or <32 weeks was more prevalent if asthma prescriptions had been discontinued before pregnancy, when compared with the whole population [aOR 1.22, 1.06–1.41, and aOR 1.53, 1.11–2.10,]. Any asthma medicine prescription before or during pregnancy was not associated with birth at <37 weeks, but was associated with extreme prematurity [aOR 1.33, 1.09–1.61]. Birth at <32 weeks was associated with OCS prescriptions [aOR 2.16, 1.07–4.37]. Birth at <37 and <32 weeks' gestation was more prevalent if only SABA had

**Table 2. Premature birth <37 weeks (n = 107,573).**

| Exposure | Exposed n [%] | Unexposed n [%] | Unadjusted OR [95% CI] | Adjusted* OR [95% CI] |
|---|---|---|---|---|
| Asthma | 783/12,690 [6.17] | 5374/94,883 [5.66] | 1.10 [1.01 to 1.18] | 1.08 [0.99 to 1.18] |
| Medicated asthma | 538/9,101 [5.91] | 5619/98,472 [5.71] | 1.04 [0.95 to 1.14] | 1.02 [0.92 to 1.12] |
| Un-medicated asthma | 245/3,589 [6.83] | 5912/103,984 [5.69] | 1.22 [1.07 to 1.39] | 1.22 [1.06 to 1.41] |
| SABA only | 251/3,820 [6.57] | 5906/103,753 [5.69] | 1.17 [1.02 to 1.33] | 1.13 [0.98 to 1.30] |
| SABA any | 506/8,413 [6.01] | 5651/99,160 [5.70] | 1.06 [0.96 to 1.16] | 1.04 [0.94 to 1.15] |
| ICS only | 12/374 [3.21] | 6145/107,199 [5.73] | 0.55 [0.31 to 0.97] | 0.64 [0.36 to 1.15] |
| ICS any | 199/3,875 [5.14] | 5958/103,698 [5.75] | 0.89 [0.79 to 1.03] | 0.87 [0.74 to 1.02] |
| LABA any | 18/465 [3.87] | 6139/107,108 [5.73] | 0.66 [0.41 to 1.06] | 0.60 [0.35 to 1.03] |
| LKA any | <5 /89 | 6155/107,484 [5.73] | OR >1Not significant | OR >1Not significant |
| OCS asthma | 30/519 [5.78] | 6127/107,054 [5.72] | 1.01 [0.70 to 1.46] | 0.93 [0.62 to 1.40] |

Exclusions from analysis: all anomalies, terminations of pregnancy for foetal anomalies [TOPFA], stillbirths, multiple births [twins, triplets and quadruplets [no higher multiples in the dataset]], exposure to insulin, anti-epileptic drugs [AEDs] or coumarins in the quarter preceding pregnancy and trimester 1, heavy drinking/substance misuse [any record]. For each analysis we compared the exposed with the unexposed population.

*adjusted for parity, smoking, year of birth, socio-economic status [SES] as Townsend fifth [quintile], and maternal age as: <20, 20–24, 25–29, 30–34, 35–39, 40–44, 45–49, 50–54, >54.

Deprivation [Townsend] scores, ranks and fifths are based on geographical area of residence, using Lower Super Output Areas [LSOAs] defined by residential postcodes. This measure of material deprivation is calculated from rates of unemployment, vehicle ownership, home ownership, and overcrowding [Townsend et al 1988 [45].

Exposures were not mutually exclusive. All those exposed to OCS, LABA and LKA were exposed to other asthma medicine5.

Abbreviations and definitions are listed in Table 1.

OR odds ratio, CI confidence intervals.

The prevalence of premature birth reported here is congruent with WHO data31

been prescribed, but interval estimates included no difference [aOR 1.13, 0.98–1.30 and 1.37, 0.99–1.89]. ICS (any or only) was associated with reduced prevalence of birth at <37 weeks', but interval estimates were compatible with no difference [aOR 0.87[0.74–1.02] and 0.64 (0.36–1.15)] [Tables 2 and 3].

## SGA

Birth weight <10th centile was associated with any asthma medication before or during pregnancy, SABA, ICS, LABA, LKA, and OCS. If prescriptions were discontinued or only ICS

**Table 3. Premature birth <32 weeks (n = 107,573).**

| Exposure | Exposed n [%] | Unexposed n [%] | unadjusted OR [95% CI] | adjusted* OR [95% CI] |
|---|---|---|---|---|
| Asthma | 131/ 12,690 [1.03] | 802/ 94,883 [0.85] | 1.22 [1.02 to 1.47] | 1.33 [1.09 to 1.61] |
| Medicated asthma | 88/9,101 [0.97] | 845/98,472 [0.86] | 1.13 [0.91 to 1.41] | 1.21 [0.96 to 1.53] |
| Un-medicated asthma | 43/3,589 [1.20] | 890/103,984 [0.86] | 1.40 [1.03 to 1.91] | 1.53 [1.11 to 2.10] |
| SABA only | 43/3,820 [1.13] | 890/103,753 [0.86] | 1.32 [0.97 to 1.79] | 1.37 [0.99 to 1.89] |
| SABA any | 84/8,413 [1.00] | 849/99,160 [0.86] | 1.17 [0.93 to 1.46] | 1.26 [0.99 to 1.59] |
| ICS only | <5/374 | 929-932/107,199 | OR <1Not significant | OR<1 Not significant |
| ICS any | 30/3,875 [0.77] | 903/103,698 [0.87] | 0.89 [0.62 to 1.28] | 0.92 [0.63 to 1.35] |
| LABA any | <5 /465 | 929/107,108 | OR <1Not significant | OR >1Not significant |
| LKA any | <5 /89 | 932/107,484 | OR >1Not significant | OR >1Not significant |
| OCS and asthma | 9/519 [1.73] | 924/107,054 [0.86] | 2.03 [1.05 to 3.93] | 2.16 [1.07 to 4.37] |

Exclusions, definitions, adjustments and abbreviations as in Table 2.

**Table 4. Small for Gestational Age [SGA] <10th centile (n = 105,331).**

| Exposure | Exposed n [%] | Unexposed n [%] | unadjusted OR [95% CI] | adjusted* OR [95% CI] |
|---|---|---|---|---|
| Asthma | 1192/12,413 [9.60] | 7965/92,918 [8.58] | 1.13 [1.06 to 1.21] | 1.10 [1.03 to 1.18] |
| Medicated asthma | 845/8,904 [9.50] | 8312/96,427 [8.62] | 1.11 [1.03 to 1.20] | 1.09 [1.01 to 1.19] |
| Un-medicated asthma | 347/3,509 [9.89] | 8810/101,822 [8.65] | 1.16 [1.04 to 1.30] | 1.10 [0.98 to 1.25] |
| SABA only | 343/3,739 [9.17] | 8814/101,592 [8.68] | 1.06 [0.95 to 1.19] | 1.02 [0.90 to 1.15] |
| SABA any | 785/8,225 [9.54] | 8372/97,106 [8.62] | 1.12 [1.04 to 1.21] | 1.09 [1.01 to 1.19] |
| ICS only | 35/370 [9.36] | 9122/104,961 [8.69] | 1.10 [0.78 to 1.56] | 1.16 [0.80 to 1.69] |
| ICS any | 380/3,786 [10.04] | 8777/101,545 [8.64] | 1.18 [1.06 to 1.31] | 1.16 [1.03 to 1.30] |
| LABA any | 57/457 [12.47] | 9100/104,874 [8.68] | 1.50 [1.14 to 1.98] | 1.75 [1.31 to 2.34] |
| LKA any | 12/87 [13.48] | 9145/105,244 [8.69] | 1.68 [0.91 to 3.10] | 1.93 [1.01 to 3.69] |
| OCS and asthma | 58/510 [11.37] | 9099/104,821 [8.68] | 1.35 [1.03 to 1.78] | 1.41 [1.05 to 1.90] |

Exclusions, definitions, adjustments and abbreviations as in Table 2

%s reported are of those with data.

–Unadjusted OR not significant

prescribed, odds ratios were similar, but interval estimates included no difference. Lower odds ratios were seen where SABA was prescribed alone. LABA and LKA were associated with birth weight <3rd centile, but numbers were low [Tables 4, 5].

## Breastfeeding

Prevalence of breastfeeding at 6–8 weeks was some 3% lower following prescriptions before or during pregnancy or prescription discontinuation, but interval estimates in adjusted analyses included no difference. Association with any SABA or any ICS in unadjusted analyses diminished when smoking, maternal age and SES were accounted [Table 6].

In all analyses, smoking and SES were significant in the final models. Maternal age predicted breastfeeding, but not preterm delivery or SGA.

## Discussion

Prescription of asthma medicines during pregnancy or in the preceding year was associated with birth before 32 weeks' gestation and at <10th centile, and exclusive formula feeding at

**Table 5. Small for Gestational Age [SGA] <3rd centile (n = 105,331).**

| Exposure | Exposed n [%] | Unexposed n [%] | unadjusted OR [95% CI] | adjusted* OR [95% CI] |
|---|---|---|---|---|
| Asthma | 260/12,413 [2.09] | 1707/92,918 [1.84] | 1.14 [1.00 to 1.30] | 1.07 [0.92 to 1.23] |
| Medicated asthma | 179/8,904 [2.01] | 1788/96,427 [1.85] | 1.09 [0.93 to 1.26] | 1.03 [0.87 to 1.22] |
| Un-medicated asthma | 81/3,509 [2.31] | 1886/101,822 [1.85] | 1.25 [1.00 to 1.57] | 1.14 [0.89 to 1.46] |
| SABA only | 78/3,739 [2.09] | 1889/101,592 [1.86] | 1.13 [0.89 to 1.41] | 1.02 [0.80 to 1.31] |
| SABA any | 164/8,225 [1.99] | 1803/97,106 [1.86] | 1.08 [0.92 to 1.26] | 1.02 [0.86 to 1.21] |
| ICS only | 5-9/(370–373) [<5]** | 1958-1962/104,961 [1.87] | 1.16 [0.58 to 2.34] | 1.22 [0.57 to 2.59] |
| ICS any | 78/3,786 [2.06] | 1889/101,545 [1.86] | 1.11 [0.88 to 1.40] | 1.09 [0.85 to 1.39] |
| LABA any | 13/457 [2.84] | 1954/104,874 [1.86] | 1.54 [0.89 to 2.68] | 1.84 [1.05 to 3.22] |
| LKA any | 5-9/(85–88) [<10]** | 1962/105,244 [1.86] | 3.21 [1.30 to 7.94] | 3.21 [1.16 to 8.93] |
| OCS asthma | 9/510 [1.76] | 1958/104,821 [1.87] | 0.94 [0.49 to 1.83] | 0.96 [0.48 to 1.29] |

Exclusions, definitions, adjustments and abbreviations as in Table 2

**These exact numbers cannot be disclosed, because these exposures are associated with <5 cases where the birth centile is unknown for this exposure.

%s reported are of those with data.

**Table 6. Breastfeeding at 6–8 weeks (n = 38,725).**

| Exposure | Exposed n [%] | Unexposed n [%] | unadjusted OR [95% CI] | adjusted* OR [95% CI] |
|---|---|---|---|---|
| Asthma | 1499/4,915 [30.50] | 11,294/33,810 [33.40] | 0.88 [0.82–0.93] | 0.93 [0.87 to 1.01] |
| Medicated asthma | 1086/3,519 [30.86] | 11,707/35,206 [33.25] | 0.90 [0.83–0.97] | 0.94 [0.87 to 1.03] |
| Un-medicated asthma | 413/1,396 [29.58] | 12,380/37,329 [33.16] | 0.85 [0.75–0.95] | 0.92 [0.81 to 1.05] |
| SABA only | 453/1,449 [31.26] | 12,340/37,276 [33.10] | 0.92 [0.82–1.03] | 1.00 [0.88 to 1.14] |
| SABA any | 1006/3,275 [30.72] | 11,787/35,450 [33.25] | 0.89 [0.82–0.96] | 0.95 [0.87 to 1.04] |
| ICS only | 35/113 [30.97] | 12,758/38,612 [33.04] | 0.91 [0.61–1.36] | 0.81 [0.52 to 1.28] |
| ICS any | 423/1,403 [30.15] | 12,370/37,322 [33.14] | 0.87 [0.78–0.98] | 0.91 [0.80 to 1.04] |
| LABA any | 52/163 [31.90] | 12,741/38,562 [33.04] | 0.95 [0.68–1.32] | 0.92 [0.62 to 1.36] |
| LKA any | 19/45 [42.22] | 12,774/38,680 [33.02] | 1.48 [0.82–2.68] | 1.73 [0.91 to 3.30] |
| OCS asthma | 71/234 [30.34] | 12,722/38,491 [33.05] | 0.88 [0.67–1.17] | 0.96 [0.70 to 1.31] |

Exclusions, definitions, adjustments and abbreviations as in Table 2

%s reported are of those with data.

6–8 weeks. However, discontinuation of medicines was associated with premature birth [<37 and <32 weeks]; whilst exclusive formula feeding and SGA were more prevalent, interval estimates were compatible with no effect in adjusted models.

Adverse outcomes were more common amongst women treated with OCS, the most severely asthmatic, but the highest prevalence of premature birth and exclusive formula feeding was amongst those discontinuing prescriptions. ICS prescription appeared to offer some protection against prematurity. Although SABA-only indicates the least severe asthma [BTS5 stage 1], women prescribed ICS for more persistent symptoms [BTS stage 2] had better outcomes [12].

## Premature birth

The association with premature birth reflects findings of a meta-analysis indicating increased risks for women diagnosed with asthma [3] or using OCS [47], contradicting earlier studies [6–8]. We did not confirm that increased risks are confined to those with more severe asthma [10, 11]. Rather, risks appeared to be increased by prescription discontinuation, which may worsen symptoms [48, 49]. Preterm birth appeared more likely where only SABA were prescribed, although interval estimates included no difference, supporting recent guidelines no longer recommending 'SABA only' regimens for any asthma [50].

## SGA

Intrauterine growth restriction is associated with asthma [8, 9, 51, 52], particularly if severe [53–55], or symptomatic [7] or accompanied by reduced spirometry parameters [10], congruent with our findings that all asthma medicines, except ICS-only or SABA-only, were associated with increased prevalence of SGA (<10th centile). Prescription discontinuation increased the prevalence of SGA, but interval estimates included no difference [55]. SGA is often secondary to placental insufficiency, subsequent to vasoconstriction of any aetiology, including pre-eclampsia, hypoxia [56, 57], smoking, cocaine [58], or, possibly, other sympathomimetics, such as beta2 agonists. Associations between LABA and birth weight <10th or <3rd centile and SABA with <10th centile may reflect vasoconstrictor properties [Tables 4, 5]. Associations with OCS, as elsewhere [7, 47], may reflect hypoxic asthma exacerbations or severity or the pharmacodynamics of corticosteroids. The associations between LABA and LKA and SGA <3rd centile may be confounded by asthma severity. These data might support a dose-

response vasoconstriction hypothesis, only partly dependent on hypoxia: infants exposed to more than one medicine, likely associated with more severe asthma, were rather more vulnerable to SGA.

## Un-medicated asthma

Many women (28.3%) discontinued their asthma medicines during pregnancy, unlike in Scandinavian cohorts [59, 60]. Un-medicated asthma was associated with premature birth, but the interval estimates for associations with SGA and exclusive formula feeding included no difference.

To meet the physiological demands of pregnancy, tidal volume and resting alveolar ventilation increase and expiratory reserve declines [61, 62]. In some women, pulmonary reserve may be insufficient to meet the increased demands for oxygen, accounting for increases in prescribing of SABA and ICS in trimester 2 [4]. In women with severe asthma, symptoms of asthma, and attendant risks of hypoxia, may intensify late in trimester 2 or *post-partum* [62]. In contrast, some women become symptom-free during pregnancy [16]. However, even if asthma becomes asymptomatic, persistent airway remodelling [basement membrane thickening] and immunological changes, predictive of subclinical bronchial hyper-responsiveness, remain [63]. Some women voluntarily suspend medication during pregnancy, opting to tolerate symptoms rather than risk harming their unborn children [64]. Any uncontrolled asthma may cause chronic or intermittent hypoxia and inflammation, which adversely affect both mother and foetus [62, 63], and may impair foetal development, causing intrauterine growth restriction, foetal distress, pre-term birth and, occasionally, death: stillbirths were more prevalent amongst women discontinuing asthma prescriptions [0.70%] when compared with UK rates 2000–2010 [0.51–0.58%] [65], and women never prescribed asthma medicines [0.38%] [S1 Table].

Non-adherence to asthma therapies exacerbates symptoms [5, 66]. Absence of prescriptions for 9 months indicates withdrawal of therapy, as it is unlikely that stockpiled inhalers would be sufficient over this length of time. We do not know why women discontinued prescriptions or whether they experienced respiratory symptoms or they or their healthcare professionals attributed dyspnea to pregnancy [67] or they remained well themselves, while their foetuses suffered episodes of hypoxia. Hypoxia stimulates release of placental corticotrophin-releasing hormone [CRH] and inflammatory cytokines. Placental CRH increases maternal and foetal cortisol production, which augments placental CRH release *via* positive feedback mechanisms, amplifying signals that may culminate in premature birth in ~5% infants [54, 68]. The strongest predictors of premature birth were medicines discontinuation and OCS, supporting the hypoxic trigger hypothesis. The apparent protection against prematurity afforded by ICS suggests that ICS may have been controlling inflammation and asthma, thereby preventing hypoxia, and consequent CRH release, despite the potential to mimic endogenous corticosteroids. Only 374 women were using ICS alone, and, since they did not need SABA, their asthma was probably well controlled, suggesting little risk from hypoxia. Beta2 agonists were formerly prescribed as tocolytics, but here they did not protect against premature birth, indicating that other factors were over-riding their actions on myometrial receptors.

## Breastfeeding

Breastfeeding rates were reduced if asthma medicines had been prescribed before or during pregnancy, and if prescriptions had been discontinued, but adjusted interval estimates included no difference [Table 6]. Assurances [5, 28] on the safety of SABA, ICS, most LABA, and OCS at <40mg per day may have contributed to the absence of a negative effect of

prescriptions on breastfeeding. Manufacturers advise against breastfeeding while using LKA; [28, 69] however, it would appear these warnings were discounted.

Beta2 agonists are generally considered safe during lactation [28, 69], although high doses and oral administration can cause tachycardia and restlessness in the infant [70]. Only small quantities of inhaled medicines reach the infant, and further work is needed to explore whether sympathomimetic effects cause sufficient restlessness [70] to impede latching, making early breastfeeding more difficult and painful or whether the sympathomimetic properties of beta2 agonists might hinder lacto genesis: [71] either might account for a reduced prevalence of breastfeeding, which was not seen here when SES, maternal age and smoking were accounted. Breastfeeding was not adversely affected by LKA and OCS, which are prescribed for severe asthma. Transfer of ICS to the infant is low [69], but too few women were prescribed only ICS to assess the impact on breastfeeding.

The breastfeeding rates at 6–8 weeks reported here reflect Wales's current data (2131/6366, 33.47%) [72]. Contemporary recruited cohorts elsewhere report higher breastfeeding rates amongst women with asthma, for example, 536/605, 88% at birth, 230/605 38% at 6 months [73]. Increasing breastfeeding rates in the UK by 0.1% would offer considerable benefit: the associated small increase in infants' average IQ gains >£27.8 million in economic productivity across each annual birth cohort [74], suggesting that the ~3% improvement identified here would be worthwhile. Breastfeeding support should not end at birth. Mild or moderate asthma may worsen *post-partum*: how this, and the associated loss of sleep, affects breastfeeding is unknown.

## Supporting women with asthma

Premature birth, SGA and suboptimal breastfeeding are public health concerns [75]. Extreme prematurity carries a 6.2% risk of cerebral palsy and a 40% risk of broncho-pulmonary dysplasia, whilst birth before 37 weeks carries a 1% risk of cerebral palsy and 40% risk of cognitive impairment [76]. Birth weight below the 10th centile is linked to a range of conditions from neonatal asphyxia to mental health conditions in adulthood [58], changes in brain morphology [77] and beta cell function [78].

The associations between un-medicated asthma and premature birth support the consensus that uncontrolled asthma increases the risk of adverse perinatal outcomes subsequent to premature birth [3], but risks were not confined to severe asthma [5, 79]. Rather, we highlight the importance of identifying the 28% of women with asthma who completely discontinue medicines, often ICS [80], as being at increased risk of premature birth and exclusive formula feeding; these women were apparently without symptoms sufficiently severe to warrant prescriptions.

We confirmed associations with SES and smoking [58, 81], but asthma medicines before or during pregnancy or their discontinuation independently predicted adverse outcomes. Lower family income and education intensify the impact on families of behavioural and neurodevelopmental sequelae of premature birth [cerebral palsy, cognitive impairment, impaired hearing or vision] [82], and the concentration of adverse outcomes and asthma prescriptions amongst the most economically deprived in Wales [83] [S1 and S2 Tables] accentuates the importance of using these findings to target support.

Prescription patterns, regardless of diagnoses, particularly discontinuation in pregnancy, offer primary care practitioners a convenient marker to identify pregnancies at increased risk, analogous to 'intention to treat' analyses: pragmatic but of unproven biological certainty. Alternative markers, such as smoking or socioeconomic deprivation, would not identify women discontinuing prescriptions [S1 and S2 Tables]. Women with severe illness are usually

actively managed in specialist clinics in secondary or tertiary care, minimizing risks reported here: [3] our findings on prescription discontinuation, SABA and 'SABA-only' support this. Although current UK guidelines do not recommend additional surveillance before or during pregnancy [5] or initiatives to target support to optimize prescription regimens and infant feeding, 'at risk' women [those discontinuing or using SABA or OCS] need extra help. Additional monitoring by specialist respiratory nurses, in parallel with midwifery care, can improve asthma control [84], but large studies are needed to assess impact on perinatal outcomes.

### Strengths and limitations

Retrospective cohorts, such as ours, have minimal volunteer selection and attrition bias [83] and no recall bias [24].

### Generalisation

The prevalence of asthma prescribing in our cohort reflects that across the UK and USA, but there is less confidence in **generalisation** to populations with lower prescription [3, 4], or higher breastfeeding rates. Coverage was limited to 40% of the population, due to suboptimal GP participation [27]. Restrictions of our cohort were due to incomplete GP participation and data, due to practices' technical failures or women moving to practices not covered by SAIL [27]. The breastfeeding outcome was further restricted by variations in recording practices across the seven Health Boards, but there was no self-selection by participants. Had a volunteer bias been operating, we should have expected to see older, less deprived women over-represented [83]; this was not the case.

Restricting the dataset by excluding pregnancies exposed to AEDs, insulin, coumarins, heavy drinking or substance misuse and multiple pregnancies increased internal validity without introducing bias, at the cost of generalisation to these groups [37, 85]. The higher prevalence of these exposures amongst women prescribed asthma medicines (S1 Table) would have caused overestimation of harm attributed to asthma medicines. Rarer co-morbidities were not excluded. Co-morbid rheumatoid arthritis and inflammatory bowel disease were more likely amongst women prescribed oral corticosteroids, who would have received specialist care [5].

### Detail in the database

We have **no information** as to whether premature births were spontaneous or induced: asthma is associated with conditions necessitating preterm induction of labour, such as pre-eclampsia [54, 86]. Neither population databases nor fieldwork fully capture symptoms, histories, spirometry, oxygen saturation, or psychosocial stressors. Maternal stress, short and long-term, increases the risk of pre-term birth [68, 81]. Substance misuse is notoriously difficult to capture, and may be under-reported; however, it is often persistent. We excluded the most vulnerable women [39–42]. The proportion of missing data [35.5% overall, 29.3% in women with asthma], precluded inclusion of BMI in regression analyses [S1 Table]. Any increased prevalence of these predictors of adverse outcomes amongst exposed women would lead to over-estimation of associations [87]. Paternity and paternal sibships are not recorded, as in many databases [88], but impact on perinatal outcomes are believed to be modest [89]. We considered each pregnancy separately, as exposures and covariates (e.g. smoking, deprivation) often vary between pregnancies. We acknowledge the genetic determinants of perinatal outcomes [90], but, without information on their association with asthma, are unable to predict how accounting for this would have affected this study. The database's timeframe did not allow us to distinguish women receiving their first ever prescription during pregnancy.

No data were available on miscarriage and induced terminations. Confining analyses to otherwise healthy live-born infants introduces a survival / survivorship bias, leaving some harms unreported, and under-estimating overall harm [87]. This was confirmed by the slightly higher rates of stillbirth (S1 Table). Congenital anomalies and stillbirths are associated with SGA and preterm birth [88], and the higher prevalence of congenital anomalies amongst women exposed to asthma medicines in this cohort is known [19]; therefore, affected infants were excluded.

We have no data on dispensing or hospital or private prescribing or internet purchasing. It is unlikely that anyone hospitalized with asthma or consulting privately would not be receiving prescriptions from primary care practitioners. Abolition of prescription charges in Wales in 2007 reduces the likelihood of private prescriptions and purchasing. Our data predate primary care prescription of monoclonal antibodies for severe eosinophilic asthma persisting despite oral corticosteroid therapy. Manufacturers advise avoiding these medicines in pregnancy [28]; therefore, population databases may be unable to describe their safety in pregnancy for some time. Too few women were prescribed LABA (465) or LKA (89) or ICS only (362) to draw conclusions relating to extreme prematurity or SGA, but findings are congruent with other studies [86].

We do not know whether any women had been over- or mis-diagnosed with asthma [91], or if any withdrawal symptoms arose [92–95] and remained un-medicated. Without data on medicines purchased without prescription, we could not explore the hypothesis that 'asthma with hay fever' is a stronger predictor of premature birth than non-atopic asthma [86]. Similarly, paucity of testing for exhaled nitric oxide, known not to improve outcomes [14], made it impossible to ascribe eosinophilic or allergic aetiology to the diagnosis.

**Adherence to prescribed regimens** cannot be ascertained from prescription or dispensing data. 30–50% of patients prescribed long-term treatment do not take medicines as recommended [92], and we also have no information on inhaler techniques or any stockpiling or sharing of inhalers. (Excluding substance misusers reduces the latter possibility.) Non-adherence dilutes the impact of medicines on outcomes; however, since discontinuation was associated with harm, unreported discontinuation in those receiving prescriptions might lead to an over-estimation of harms attributed to prescriptions.

## Multiple testing

We acknowledge the hazards of **multiple testing**, without correction. Our independent (predictor) variables are highly correlated. This makes standard adjustments, such as the Bonferroni method, unduly conservative, risking false negatives [96]. Observation studies are limited by unmeasured confounding, and the impossibility of including all contextual variables [97, 98]. Our findings are consistent with biological plausibility and the literature.

## Confounding and effect mediation

For most adverse outcomes, retrospective analysis yields lower odds or risk ratios than prospective studies [93]. However, this is not true of cohorts examining asthma in pregnancy. This might be interpreted as prospective cohort studies offering enhanced opportunities to control asthma, particularly to optimize prophylaxis with ICS [3], supported here by suggestions of improved outcomes with ICS prescriptions.

In evaluating the impact of medicines using population databases, it is rarely possible to avoid residual **confounding by indication** and severity of indication without randomisation [99, 100]. Prescriptions are sometimes taken as proxies for diagnoses [101], as diagnoses are not always recorded in healthcare databases, and these medicines have no other indications in

the UK [28]. Since the risks of poorly controlled asthma outweigh the risks of harm to the foetus, only the mildest maternal asthma is un-medicated on medical advice [5, 50], but we do not know whether continuing prescriptions indicated recurrent symptoms or a propensity to compliance. We acknowledge the difficulties of disentangling the effects of asthma from its management. Nevertheless, these findings should alert practitioners to the need for targeted care preconception, and during pregnancy and lactation.

### Implications: Support before, during and after pregnancy

The clinical consequences of premature birth, including cerebral palsy and cognitive impairment [76], increase the importance of the modest [ORs <2] associations identified. Women discontinuing prescriptions were probably relatively unaffected by symptoms, whilst an additional ~1% of their infants were born preterm: discontinuation may have increased the risk of hypoxia [and hence premature birth]. Taken with lower breastfeeding rates, these findings identify unmet needs amongst women discontinuing prescriptions, and militate against medication reduction in pregnancy [102]. Reduced risks with ICS suggest that increased monitoring, targeted support and active asthma management [3, 50] are needed before, during and after pregnancy. Programming primary care electronic records to alert professionals to contact women who leave >4 months between prescriptions for asthma medicines might make this feasible.

### Supporting information

**S1 Checklist.**
(DOC)

**S1 Table. Demographic details of population, excluding congenital anomalies [n = 113316].**
(DOCX)

**S2 Table. Demographic details of subjects included in the analysis [n = 107573].**
(DOCX)

**S3 Table. Medicated and unmedicated asthma: Descriptive data n = 107573.**
(DOCX)

### Acknowledgments

This study uses anonymised data held in the Secure Anonymised Information Linkage [SAIL] system, which is part of the national e-health records research infrastructure for Wales. We should like to acknowledge all the data providers who make anonymised data available for research. Data held in SAIL databases are anonymised and aggregated and have been obtained with permission of relevant Data Protection Officers, as approved by the National Research Ethics Service, Wales. Permissions are conditional on non-disclosure of numbers <5 in any cell. The project was approved by the SAIL Information Governance Review Panel [IGFRP] on 24th March 2011. Since the project uses only anonymised data, ethical review was deemed unnecessary.

   **We should like to thank**: Hildrum Sundseth from the European Institute of Women's Health for advice as service user representative on EUROmediCAT, and Professor Helen Dolk for her role as lead of EUROmediCAT and continuing support.

## Author Contributions

**Conceptualization:** Gareth Davies, Sue Jordan, David Tucker.

**Data curation:** Gareth Davies, Sue Jordan, David Tucker.

**Formal analysis:** Gareth Davies, Sue Jordan.

**Funding acquisition:** Sue Jordan, David Tucker.

**Investigation:** Gareth Davies, Sue Jordan, David Tucker.

**Methodology:** Gareth Davies, Sue Jordan.

**Project administration:** Gareth Davies, Sue Jordan.

**Resources:** Sue Jordan.

**Visualization:** Gareth Davies, Sue Jordan, Daniel Thayer, David Tucker, Ioan Humphreys.

**Writing – original draft:** Sue Jordan.

**Writing – review & editing:** Gareth Davies, Sue Jordan, Daniel Thayer, David Tucker, Ioan Humphreys.

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
