## [Decision Letter · Decision Letter 0]

29 Apr 2020

PONE-D-20-02812

Medicines prescribed for asthma, discontinuation and perinatal outcomes, including breastfeeding: a population cohort analysis

PLOS ONE

Dear Professor Jordan,

Thank you for submitting your manuscript to PLOS ONE. After careful consideration, we feel that it has merit but does not fully meet PLOS ONE’s publication criteria as it currently stands. Therefore, we invite you to submit a revised version of the manuscript that addresses the points raised during the review process.

Please pay particular attention to concerns shared by the reviewers: the potential selection bias introduced by excluding  stillbirths and severe congenital malformations; the conflations of disease severity and medication type; the need to clarify why asthma medication  discontinuation in pregnancy might be problematic or important; clarifying the comparisons/reference groups.

Please also note reviewer 1's concerns about a causal interpretation. If you, the authors are not committed to a causal interpretation, then a different justification for the study  will need to be provided.        

We would appreciate receiving your revised manuscript by Jun 13 2020 11:59PM. To enhance the reproducibility of your results, we recommend that if applicable you deposit your laboratory protocols in protocols.io, where a protocol can be assigned its own identifier (DOI) such that it can be cited independently in the future. For instructions see: http://journals.plos.org/plosone/s/submission-guidelines#loc-laboratory-protocols

We look forward to receiving your revised manuscript.

Kind regards,

Abigail Fraser

Academic Editor

PLOS ONE

Journal Requirements:

2. We noted in your submission details that a portion of your manuscript may have been presented or published elsewhere:

'Antidepressant prescriptions, discontinuation, depression and perinatal outcomes, including breastfeeding: A population cohort analysis (2019) Sue Jordan, Gareth I. Davies, Daniel S. Thayer David Tucker, Ioan Humphreys '

Please clarify whether this publication was peer-reviewed and formally published. If this work was previously peer-reviewed and published, in the cover letter please provide the reason that this work does not constitute dual publication and should be included in the current manuscript.

Reviewers' comments:

Reviewer's Responses to Questions

**Comments to the Author**

1. Is the manuscript technically sound, and do the data support the conclusions?

Reviewer #1: Partly

Reviewer #2: Partly

Reviewer #3: Yes

Reviewer #4: Yes

2. Has the statistical analysis been performed appropriately and rigorously? 

Reviewer #1: Yes

Reviewer #2: I Don't Know

Reviewer #3: Yes

Reviewer #4: I Don't Know

3. Have the authors made all data underlying the findings in their manuscript fully available?

Reviewer #1: Yes

Reviewer #2: Yes

Reviewer #3: Yes

Reviewer #4: Yes

4. Is the manuscript presented in an intelligible fashion and written in standard English?

Reviewer #1: Yes

Reviewer #2: Yes

Reviewer #3: Yes

Reviewer #4: Yes

5. Review Comments to the Author

Reviewer #1: The authors used a secure anonymous information linkage from wales to examine whether women who use asthma medication have an increased risk of adverse pregnancy outcomes. They report an increased risk of gestational age <32 weeks and small-for-gestational age (SGA) among women who use asthma medications during pregnancy, in addition to a lower likelihood of breastfeeding. Women who discontinued their use of asthma medications during pregnancy were found to have a particularly increased risk of preterm birth and SGA.

1) The authors chose to exclude all stillbirths and all live births with malformations. These are other pregnancy outcomes that could be on the causal pathway for the relationship between use of asthma medications during pregnancy and the adverse pregnancy outcomes the authors are studying. It would be good if the authors could better explain their rational, particularly due to the fact that these exclusions could have introduced bias (specifically collider bias).

2) Did any women contribute to the analysis with more than one pregnancy ? If so, how was this accounted for?

3) I do not understand why the authors chose to not account for maternal age. Their indirect claim that this additional adjustment would not influence the associations seems insufficient without presenting how the results changed with this additional adjustment.

4) On the one hand, the authors say clearly that what they are presenting are associations and not causal effects, but on the other hand they choose to present the number needed to harm (NHH). Based on their limited opportunity to account for potential confounding factors, and the high likelihood of unmeasured confounding, I strongly recommend that they take these estimations out of the paper. The authors should also state that their lack of information on BMI is an important limitation of the study.

5) Can the authors please confirm that they did not have any information available on other allergic diseases that would allow them to try to distinguish between allergic and non-allergic asthma?

6) It would be great if the authors could make it even more clear how the different medications might reflect the severity of the underlying asthma symptoms. How can the results across the different medications groups be interpreted as reflecting disease severity?

7) I think that the authors need to be careful in drawing the conclusion that discontinuation of asthma medications during pregnancy is particularly harmful. They do not know whether these women had mild symptoms, nor indeed how well any of the women were regulated due to their lack of information on asthma symptoms. I therefore consider their second sentence in the conclusion to be an overstatement (“Identifying women discontinuing asthma medicines during pregnancy from prescription records and offering close monitoring and support for breastfeeding might address these problems.”). This sentence should be deleted and the discussion appropriately revised.

8) Would it be possible to conduct a comparison of the risk of adverse pregnancy outcomes and the likelihood of breastfeeding across the groups of or combinations of asthma medications? The reference group is now those who do not use asthma medications. It could be interesting to compare the risk between women using different asthma medications directly, to see if any of the specific medications pose a particularly high risk.

9) Are the authors able to identify the subgroup of women first diagnosed with asthma during pregnancy? There is literature indicating that this group of women might have a particularly high risk of adverse pregnancy outcomes.

Minor comments

1) It is necessary that a bit more information about the databases is presented in the methods to allow readers to fully appreciate the study population.

Reviewer #2: Thank you for giving me the opportunity to review this article, which aimed to explore associations between medication (and discontinuation of) use for asthma in pregnancy and pregnancy outcomes

This population based cohort study demonstrated an association between medication use for asthma and preterm birth(<32 weeks), and SGA, whereas discontinuation of the medications was associated with preterm birth (both <37 weeks and 32 weeks ) but not SGA.

I believe the aim of the study is important and it has been conducted well and rigorously , however I believe the reasons for non medication use, and discerning this from asthma disease severity are not well described in the article and therefore needs more clarity in the discussions and conclusions.

More detailed comments on the article;

Introduction

I believe the introduction is missing the essential concept that active poorly uncontrolled asthma is associated with preterm birth and SGA, and that medication use, especially multiple medication use is likely reflective of more poorly controlled disease. The impact of the actual medication itself on such outcomes is a separate influencing factor.

Page 4 line 37 – congenital anomalies are rare, was this a main aim of the study, and major congenital abnormalities were excluded?

Methods

Page 4 line 40 – suggest moving this sentence to the discussion

Page 4 line 49- is there a bias in the 60% of practices that did not share the data?

Page 7 line 114 – this line should be removed

Page 7 – line 118 – why exclude stillbirth

Table 2 Premature birth

Discussion

Overall discussion needs to be more concise

Page 17 line 239 – I am unsure this statement is entirely correct, women may discontinue medication in pregnancy f they have minimal symptoms but they may also discontinue medication if they are concerned about the impact of the medications on the pregnancy, this in turn can cause uncontrolled asthma with worsening severity and potentially worse pregnancy outcomes in itself.

Page 17 – line 252, which can in turn cause worsening disease

Page 17 line 258 – see above statement for line 239

Page 18 – 265- how can you separate these effects from disease severity

Page 18 line 266 – this section regarding non-adherence needs to be concisely described in the introduction

Reviewer #3: This is a large population based study, on the association between exposure to asthma medication, its discontinuation, and pregnancy outcomes, including breastfeeding. This is a relatively prevalent exposure, and the question important for healthcare providers and women planning a pregnancy. The researchers were able to include data on breastfeeding at age 6-8 weeks, which is another public health concern, and to clarify whether it is affected by maternal medication status and possible knowledge on this issue.

This is an important subject, the manuscript is well written, and the Methods and Results are clear and seem appropriate for this question. I do have a few suggestions, that need clarification or additions to the manuscript:

1. If I understand correctly, pregnancy terminations were excluded from the study, does this include stillbirths? If so, it is possible the worse outcomes were excluded from the study, and the live births included in this study are actually present a survival bias. Again, if this is the case, this possible survival bias, and it`s effect on the results needs to be addressed in the Discussion section.

2. Were there siblings in the cohort? How were they addressed, as they are not independent one from another. There are statistical methods to adjust for the similarity among the siblings, and they can be clustered by the mothers.

3. Are any of these medication prescribed for other indications? Were there mothers in the cohort that received similar medication or discontinued these medications (but not for asthma)? This may have led to a misclassification of the studied exposure. If so, it would be interesting to study differences between the defined study groups and this group of women, in terms of the pregnancy outcomes etc. This could stress out whether the association was more likely with the medication/its` discontinuation, or the asthma itself. It would also be interesting to exclude such a group from the "unexposed" group in the study, and see how the results are affected.

4. If the data is available, it would be interesting to model and compare women based on dosage of medication, or to compare women with no asthma/medication (reference group), and the following groups: women with untreated asthma, women discontinuing the medication, women receiving one medication, women receiving two medications, more than two medications. A dose response effect, if found, could strengthen the findings.

Few minor issues:

-line 4 (introduction) missing the word : Approximately 9% of pregnant women…

-line 21- SGA has been defined in previous paragraph.

-line 174, it is unclear what the comparison group was. The unexposed? Pregnancies with ongoing asthma medication?

- line 294- missing a ].

-line 318- unclear sentence, please rephrase.

-changes in font size and style throughout the manuscript, as well as spacing between lines and paragraphs.

Reviewer #4: This population-based study utilised maternal primary care data linked with data from >100,000 infants in Wales. The use and discontinuation of asthma medications for asthma in pregnancy was examined as the exposure, and adverse pregnancy outcomes and breastfeeding rates were examined.

This study adds additional information to our knowledge of the effects of asthma medication use on pregnancy outcomes, which is important given that there are very few RCTs on asthma medication use in pregnancy. Previous studies have not addressed the association of asthma or medication use for asthma and breastfeeding, an important novel aspect of this paper. However, breastfeeding information, while novel, was limited by a much smaller sample size (late introduction of data collection) and the available data was only able to be classified as any breastfeeding vs no breastfeeding.

The authors report in the introduction that they were unable to locate reports on the prevalence of breastfeeding amongst women with asthma. A recent publication in Pediatric Pulmonology (Harvey SM et al: Maternal asthma, breastfeeding and respiratory outcomes in the first year of life) contains this data. There are likely to be a few other papers from birth cohorts of infants at risk of asthma due to maternal asthma/atopy which may contain this information also. It would be helpful if the authors outlined in more detail the gap in this literature, including the recent paper from Harvey et al.

Since the paper addresses the discontinuation of asthma medicine use in pregnancy, it would be good to see a paragraph in the introduction which outlines the extent of this clinical problem in pregnant women with asthma.

The authors state that “prescriptions reflect physicians’ assessment of asthma severity” (page 11, line 67). However, I am unsure if this assumption can be made for a pregnant population. Changes in asthma medication use specific to pregnancy would suggest that some physicians change prescribing behaviour in response to pregnancy itself. This needs to be addressed (as suggested above) in the introduction.

Were LABA prescribed along with ICS, or as monotherapy? It wasn’t clear from table 1.

There was a high proportion of women who received SABA only (45.4%). Do the authors have any comments on this in the context of recent changes to the GINA guidelines (people with mild asthma should not receive SABA alone)?

The text on page 14, line 166-168 gives the numbers of women receiving ICS, LABA and LKA, and OCS – the proportions should also be added here.

Were the women who discontinued medication the 3589 (28.3%) who received their last prescription before pregnancy (line 165)? It was not clear to me. Or was there another group who discontinued medications part way through pregnancy?

The authors report that discontinuation of asthma medication before pregnancy was associated with preterm birth, but not with SGA. What is the proposed mechanism for these differing effects?

Breastfeeding prevalence at 6-8 weeks was lower in women with asthma medication before, during pregnancy, or discontinued in pregnancy. No further details were supplied in the text. This could be added, as table 6 only indicates a significant effect of asthma and unmedicated asthma in the adjusted odds ratios.

As the findings of this study focus on discontinuation of asthma medication, much more background information is needed on this issue, and more specifics on this are required in the methods/ results section.

The discussion lacks critical review of the existing literature and is difficult to follow at times.

The section on breastfeeding (line 303), describes these results as “neutral”; however, this was not the case as outlined in the results.

6. PLOS authors have the option to publish the peer review history of their article (what does this mean?). If published, this will include your full peer review and any attached files.

Reviewer #1: No

Reviewer #2: No

Reviewer #3: No

Reviewer #4: No

---

## [Author Response · Author response to Decision Letter 0]

25 Jun 2020

Medicines prescribed for asthma, discontinuation and perinatal outcomes, including breastfeeding: a population cohort analysis

Short title: Asthma medicines, discontinuation and perinatal outcomes, including breastfeeding

Gareth Davies2, Sue Jordan 1*, Gareth Davies2, Daniel Thayer2, David Tucker3, Ioan Humphreys1

Abigail Fraser

Academic Editor

PLOS ONE

Dear Dr. Fraser,

We are returning our paper, having addressed the comments of the 4 reviewers and editor, below. As the reviewers indicate, this is the first report of the impact of asthma medicines on breastfeeding, and we have important data on medicines discontinuation. We are particularly grateful where reviewers have suggested particular references. At times, we have been unable to identify literature recommended. We should, therefore be very pleased were the reviewers to find the time to indicate which articles they feel should be referenced.

As we indicated, we were invited to resubmit this paper last year, when we addressed the concerns of the 2 earlier referees. Our responses to these are appended below. A paper on antidepressants from this cohort was submitted as a sister paper to this, and was published last year in Plos One. The overlap is in the cohort, and we have added this information to the paper. 

We should like to thank you and the reviewers for their time spent on our work. Please let us know if further changes are needed.

Yours, 

Sue Jordan

Yr Athro Professor 

Coleg y Gwyddorau Dynol a Iechyd College of Human and Health Sciences

Prifysgol Abertawe Swansea University

Abertawe SA2 8PP Swansea SA2 8PP

01792 518541 01792 518541 

Cynghorydd Cymuned Gwaun Cae Gurwen Community Councillor, Gwaun Cae Gurwen

Llywydd (ar y cyd) Cangen Undeb y Brifysgol a'r Coleg Branch President (joint) UCU 

project website: http://www.swansea.ac.uk/adre/

Editorial Office

We noted in your submission details that a portion of your manuscript may have been presented or published elsewhere:

'Antidepressant prescriptions, discontinuation, depression and perinatal outcomes, including breastfeeding: A population cohort analysis (2019) Sue Jordan, Gareth I. Davies, Daniel S. Thayer David Tucker, Ioan Humphreys '

Please clarify whether this publication was peer-reviewed and formally published. If this work was previously peer-reviewed and published, in the cover letter please provide the reason that this work does not constitute dual publication and should be included in the current manuscript.

We noted this publication in Plos One in our covering letter, below, and our original submission to Plos One in 2019. The paper was peer-reviewed. We explain that the 2 papers emanate from the same cohort. Therefore, the study flow diagram, and the setting and participants are the same

Jordan S, Davies GI, Thayer DS, Tucker D, Humphreys I (2019) Antidepressant prescriptions, discontinuation, depression and perinatal outcomes, including breastfeeding: A population cohort analysis. PLOS ONE 14(11): e0225133. https://doi.org/10.1371/journal.pone.0225133

We have added to p.5:

The impact of antidepressants on breastfeeding, premature birth and SGA in this cohort has been reported [28, Jordan et al 2019].

Editor

Please pay particular attention to concerns shared by the reviewers: the potential selection bias introduced by excluding stillbirths and severe congenital malformations; 

We have added to limitations p.22

No data were available on miscarriage and induced terminations. Confining analyses to otherwise healthy live-born infants introduces a survival / survivorship bias, leaving some harms unreported (Schnitzer & Blais 2018). This was confirmed by the slightly higher rates of stillbirth (S1 file, Table A1). Congenital anomalies and stillbirths are associated with SGA and preterm birth [Miquel-Verges et al 2015]; therefore, affected infants were excluded. 

the conflations of disease severity and medication type; we added on p.18

The associations between LABA and LKA and SGA <3rd centile may be confounded by asthma severity. These data might support a dose-response vasoconstriction hypothesis, only partly dependent on hypoxia: infants exposed to more than one medicine, likely associated with more severe asthma, were rather more vulnerable to SGA. 

And on p.24

In evaluating the impact of medicines using population databases, it is rarely possible to avoid residual confounding by indication and severity of indicationwithout randomisation [85,86]

the need to clarify why asthma medication discontinuation in pregnancy might be problematic or important; 

we have added to the introduction:

Up to 50% pregnant women discontinue asthma medicines, often without professional advice, frequently worsening asthma and outcomes [1,9,14]. 

And discuss on p.19 

The strongest predictors of premature birth were medicines discontinuation and OCS, supporting the hypothesis of a hypoxic trigger. Et seq

clarifying the comparisons/reference groups.

We have written, p.8: For each analysis we compared the exposed with the unexposed population. This allows different medicines to be compared. 

Please also note reviewer 1's concerns about a causal interpretation. If you, the authors are not committed to a causal interpretation, then a different justification for the study will need to be provided. 

Our aim is to identify how best to target support, p.4:

To identify women likely to benefit from additional care, and target support, we aimed to explore any associations between prescriptions for asthma medicines during pregnancy, or their discontinuation, and the range of perinatal outcomes that matter to women [prematurity, SGA, and, for the first time, breastfeeding at 6-8 weeks, full or partial]. 

We feel that we have achieved this objective:

Prescription patterns, regardless of diagnoses, particularly discontinuation in pregnancy, offer primary care practitioners a convenient marker to identify pregnancies at increased risk, analogous to ‘intention to treat’ analyses: pragmatic but of unproven biological certainty. P.21

Taken with lower breastfeeding rates, these findings identify unmet needs amongst women discontinuing prescriptions, and militate against medication reduction in pregnancy [87]. Reduced risks with ICS suggest that increased monitoring, targeted support and active asthma management [3, 87, GINA 2020] are needed before, during and after pregnancy. Programming primary care electronic records to alert professionals to contact women who leave >4 months between prescriptions for asthma medicines might make this feasible. P.24

If this could be published, we can present the evidence to the all-Wales committees to ask for this to be done. 

Reviewer #1: The authors used a secure anonymous information linkage from wales to examine whether women who use asthma medication have an increased risk of adverse pregnancy outcomes. They report an increased risk of gestational age <32 weeks and small-for-gestational age (SGA) among women who use asthma medications during pregnancy, in addition to a lower likelihood of breastfeeding. Women who discontinued their use of asthma medications during pregnancy were found to have a particularly increased risk of preterm birth and SGA.

1) The authors chose to exclude all stillbirths and all live births with malformations. These are other pregnancy outcomes that could be on the causal pathway for the relationship between use of asthma medications during pregnancy and the adverse pregnancy outcomes the authors are studying. It would be good if the authors could better explain their rational, particularly due to the fact that these exclusions could have introduced bias (specifically collider bias). 

Thank you. We have added to p.5:

Infants with congenital anomalies were excluded, as associations with asthma prescriptions are reported elsewhere. [19]

Reporting twice on the same cases jeopardises any future meta-analyses. 

And to limitations p.22

No data were available on miscarriage and induced terminations. Confining analyses to otherwise healthy live-born infants introduces a survival / survivorship bias, leaving some harms unreported (Schnitzer & Blais 2018). This was confirmed by the slightly higher rates of stillbirth (S1 file, Table A1). Congenital anomalies and stillbirths are associated with SGA and preterm birth [Miquel-Verges et al 2015]; therefore, affected infants were excluded. 

Schnitzer ME, Blais L. Methods for the assessment of selection bias in drug safety during pregnancy studies using electronic medical data. Pharmacol Res Perspect. 2018;6(5):e00426. Published 2018 Sep 21. doi:10.1002/prp2.426 https://www.ncbi.nlm.nih.gov/pmc/articles/PMC6149369/

Miquel-Verges F, Mosley BS, Block AS, Hobbs CA. A spectrum project: preterm birth and small-for-gestational age among infants with birth defects. J Perinatol. 2015 Mar;35[3]:198-203. doi: 10.1038/jp.2014.180. Epub 2014 Oct 2. PubMed PMID: 25275696.

2) Did any women contribute to the analysis with more than one pregnancy ? If so, how was this accounted for?

Thank you. We have added to limitations p.22: Information on paternity or paternal sibships is incomplete, as in many databases. We considered each pregnancy separately, as exposures and covariates (e.g. smoking, deprivation) often vary between pregnancies. 

We accounted for parity in the analysis. 

3) I do not understand why the authors chose to not account for maternal age. Their indirect claim that this additional adjustment would not influence the associations seems insufficient without presenting how the results changed with this additional adjustment.

Thank you. None of the other 5 reviewers have made this request. Age is not a modifiable risk factor. The association with deprivation would adversely affect the collinearity of the regression models, as explained, p.7 line 131 et seq:

Maternal age, a non-modifiable risk factor, is strongly associated with SES and parity in this cohort, and there were too few women aged >39 at birth to account for this potentially high risk group separately. [42] Therefore, in view of the low numbers in some outcomes, to reduce collinearity with SES and parity, age was not entered into the regression analyses.

 We therefore focussed on the modifiable risk factors, including deprivation. We have added this to our ‘limitations’ section. We acknowledge that not including data on age detracts from the regression models, but parsimony is important where there are low numbers of exposed cases. We have added, with supporting references:

We omitted age from the models to reduce collinearity and maintain parsimony, as numbers of exposed cases were low. Observation studies are limited by unmeasured confounding, and the impossibility of including all contextual variables (York 2012, Greenland et al 2016); we opted to retain deprivation as a key policy informant.

Greenland S, Daniel R, Pearce N. Outcome modelling strategies in epidemiology: traditional methods and basic alternatives. Int J Epidemiol. 2016;45(2):565‐575. doi:10.1093/ije/dyw040

 York R. Residualization is not the answer: Rethinking how to address multicollinearity. Soc Sci Res. 2012;41(6):1379‐1386. doi:10.1016/j.ssresearch.2012.05.014

4) On the one hand, the authors say clearly that what they are presenting are associations and not causal effects, but on the other hand they choose to present the number needed to harm (NHH). Based on their limited opportunity to account for potential confounding factors, and the high likelihood of unmeasured confounding, I strongly recommend that they take these estimations out of the paper. 

Thank you. We await the editors’ advice. These calculations of the absolute risk differences are recommended in some journals and are compulsory in others, as they offer greater clarity than odds ratios. NNH conveys absolute risks, which are important to women. It is presented in the sister paper in this journal (Jordan et al 2019). None of the 5 other reviewers have asked for removal of these calculations. We should prefer to allow readers to judge the utility of the parameter for themselves. We have added the rubric, below (p.8):

To report absolute risk differences, which are important to women, the number needed to harm [NNH] was calculated for statistically significant results.

The authors should also state that their lack of information on BMI is an important limitation of the study.

We agree. The paucity of BMI data was noted as a limitation p.22: The proportion of missing data, [35.5% overall, 29.3% in women with asthma], precluded inclusion of BMI in regression analyses [S1 file, Table A1]. 

5) Can the authors please confirm that they did not have any information available on other allergic diseases that would allow them to try to distinguish between allergic and non-allergic asthma?

Thank you, we had written (Lines 422 et seq)

Without data on medicines purchased without prescription, we could not explore the hypothesis that ‘asthma with hayfever’ is a stronger predictor of premature birth than non-atopic asthma [83]. 

We have now added: Similarly, paucity of testing for exhaled nitric oxide, which may not improve outcomes [14], made it impossible to ascribe eosinophilic or allergic aetiology to the diagnosis. 

6) It would be great if the authors could make it even more clear how the different medications might reflect the severity of the underlying asthma symptoms. How can the results across the different medications groups be interpreted as reflecting disease severity?

Thank you. We have added p.5: 

Prescriptions reflect physicians’ assessment of asthma severity [5, 28], since prescribers adhere to guidelines [5] that direct monitoring and prescribing.

The notes to table 1 detail the UK prescribing guidelines, which are referenced. 

7) I think that the authors need to be careful in drawing the conclusion that discontinuation of asthma medications during pregnancy is particularly harmful. They do not know whether these women had mild symptoms, nor indeed how well any of the women were regulated due to their lack of information on asthma symptoms. I therefore consider their second sentence in the conclusion to be an overstatement (“Identifying women discontinuing asthma medicines during pregnancy from prescription records and offering close monitoring and support for breastfeeding might address these problems.”). This sentence should be deleted and the discussion appropriately revised.

Thank you, we have redrafted: Women discontinuing medicines during pregnancy could be identified from prescription records. The impact of targeting close monitoring and breastfeeding support warrants exploration. 

8) Would it be possible to conduct a comparison of the risk of adverse pregnancy outcomes and the likelihood of breastfeeding across the groups of or combinations of asthma medications? The reference group is now those who do not use asthma medications. It could be interesting to compare the risk between women using different asthma medications directly, to see if any of the specific medications pose a particularly high risk.

Thank you. We have now clarified in the notes to Table 1 that some medicines (LABA, LKA, OCS) are not prescribed alone. We have distinguished where ICS and SABA were prescribed alone. We have written, p.8: For each analysis we compared the exposed with the unexposed population. This allows different medicines to be compared, as requested. 

9) Are the authors able to identify the subgroup of women first diagnosed with asthma during pregnancy? There is literature indicating that this group of women might have a particularly high risk of adverse pregnancy outcomes.

Thank you. Unfortunately, we did not identify women with an earlier diagnosis, because the database only extends to 2000, missing any childhood asthma. Women diagnosed with asthma for the first time during pregnancy are likely to have had earlier mild symptoms overlooked. We have added to limitations: 

The timeframe did not allow us to distinguish women receiving their first ever prescription during pregnancy. 

Minor comments

1) It is necessary that a bit more information about the databases is presented in the methods to allow readers to fully appreciate the study population.

We agree, but word limit is a problem, and there are several papers on this, references 25 & 26. I’m not sure as to the exact information required, so we’ve added the link to the SAIL FAQs https://saildatabank.com/faq/. We fully describe SAIL in our EUROmediCAT report, reference 27. We have added weblinks to data sources. 

Reviewer #2: Thank you for giving me the opportunity to review this article, which aimed to explore associations between medication (and discontinuation of) use for asthma in pregnancy and pregnancy outcomes

This population based cohort study demonstrated an association between medication use for asthma and preterm birth(<32 weeks), and SGA, whereas discontinuation of the medications was associated with preterm birth (both <37 weeks and 32 weeks ) but not SGA.

I believe the aim of the study is important and it has been conducted well and rigorously , 

Thank you

however I believe the reasons for non medication use, and discerning this from asthma disease severity are not well described in the article and therefore needs more clarity in the discussions and conclusions.

Thank you. We agree, but have no information as to why medicines were discontinued or if symptoms arose. We have added to limitations p.18:

We do not know why women discontinued prescriptions or whether they suffered respiratory symptoms or they or their healthcare professionals attributed dyspnoea to pregnancy [61] or they remained well themselves, while their foetuses may have suffered episodes of hypoxia. 

More detailed comments on the article;

Introduction

I believe the introduction is missing the essential concept that active poorly uncontrolled asthma is associated with preterm birth and SGA, and that medication use, especially multiple medication use is likely reflective of more poorly controlled disease. The impact of the actual medication itself on such outcomes is a separate influencing factor.

Thank you, we agree. We have moved from the discussion to introduction p.3:

uncontrolled asthma increases the risk of adverse perinatal outcomes, [3]

We continue, duly modified: we aimed to explore any associations between prescriptions for asthma medicines during pregnancy, or their discontinuation

under ‘exposure p.5 we acknowledge:

Prescriptions reflect physicians’ assessment of asthma severity [5, 28].

And give details in Table 1 and its footnotes. 

Page 4 line 37 – congenital anomalies are rare, was this a main aim of the study, and major congenital abnormalities were excluded?

We explain p.5: 

Infants with congenital anomalies were excluded, as associations with asthma prescriptions are reported elsewhere. [19]

Methods

Page 4 line 40 – suggest moving this sentence to the discussion

Thank you, done. 

Page 4 line 49- is there a bias in the 60% of practices that did not share the data?

Thank you, yes, there was a small difference, reported in reference 27. We have now added to p.4: 

Women in the included population were slightly less deprived and older than the rest of Wales [27].

Page 7 line 114 – this line should be removed

Done 

Page 7 – line 118 – why exclude stillbirth

Congenital anomalies in these cohorts were reported elsewhere (reference 19). We should have clarified this on p.5, and have now added.

Infants with congenital anomalies were excluded, as associations with asthma prescriptions are reported elsewhere. [19]

Table 2 Premature birth

corrected – thank you

Discussion

Overall discussion needs to be more concise

We have reviewed and reduced, but needed to address reviewers’ very valid points. We addressed each outcome, the issue of unmedicated asthma and the limitations of observational work. The latter have been expanded to address the thoughtful contributions of the 6 reviewers. 

Page 17 line 239 – I am unsure this statement is entirely correct, women may discontinue medication in pregnancy f they have minimal symptoms but they may also discontinue medication if they are concerned about the impact of the medications on the pregnancy, this in turn can cause uncontrolled asthma with worsening severity and potentially worse pregnancy outcomes in itself.

We agree. This sentence has been removed.

Page 17 – line 252, which can in turn cause worsening disease

Thank you, added.

Page 17 line 258 – see above statement for line 239

Thank you, this sentence has been deleted.

Page 18 – 265- how can you separate these effects from disease severity

We agree, there is a high risk of confounding by severity here: we only suggest further work to explore. We have added:

The associations between LABA and LKA and SGA <3rd centile may be confounded by the severity of asthma, but warrant further investigation.

Page 18 line 266 – this section regarding non-adherence needs to be concisely described in the introduction

Thank you. We have moved from the discussion to introduction p.3:

uncontrolled asthma increases the risk of adverse perinatal outcomes, [3] (…) and added

Up to 50% pregnant women discontinue asthma medicines, often without professional advice, frequently worsening asthma and outcomes [1,9,14]. 

Many journals ask authors to confine the introduction to 500 words, but do not restrict the discussion. Unfortunately, we have no information on adherence, and this is included with the other limitations.

Reviewer #3: This is a large population based study, on the association between exposure to asthma medication, its discontinuation, and pregnancy outcomes, including breastfeeding. This is a relatively prevalent exposure, and the question important for healthcare providers and women planning a pregnancy. The researchers were able to include data on breastfeeding at age 6-8 weeks, which is another public health concern, and to clarify whether it is affected by maternal medication status and possible knowledge on this issue.

This is an important subject, the manuscript is well written, and the Methods and Results are clear and seem appropriate for this question. 

Thank you

I do have a few suggestions, that need clarification or additions to the manuscript:

1. If I understand correctly, pregnancy terminations were excluded from the study, does this include stillbirths? If so, it is possible the worse outcomes were excluded from the study, and the live births included in this study are actually present a survival bias. Again, if this is the case, this possible survival bias, and it`s effect on the results needs to be addressed in the Discussion section.

Thank you. Yes, this is correct. The numbers of stillbirths are reported in S1 file, Table A1, and commented in the discussion p.18. We have added to limitations p.22:

No data were available on miscarriage and induced terminations. Confining analyses to otherwise healthy live-born infants introduces a survival / survivorship bias, leaving some harms unreported (Schnitzer & Blais 2018). This was confirmed by the slightly higher rates of stillbirth (S1 file, Table A1). Congenital anomalies and stillbirths are associated with SGA and preterm birth [Miquel-Verges et al 2015]; therefore, affected infants were excluded. 

Schnitzer ME, Blais L. Methods for the assessment of selection bias in drug safety during pregnancy studies using electronic medical data. Pharmacol Res Perspect. 2018;6(5):e00426. Published 2018 Sep 21. doi:10.1002/prp2.426 https://www.ncbi.nlm.nih.gov/pmc/articles/PMC6149369/

Miquel-Verges F, Mosley BS, Block AS, Hobbs CA. A spectrum project: preterm birth and small-for-gestational age among infants with birth defects. J Perinatol. 2015 Mar;35[3]:198-203. doi: 10.1038/jp.2014.180. Epub 2014 Oct 2. PubMed PMID: 25275696.

2. Were there siblings in the cohort? How were they addressed, as they are not independent one from another. There are statistical methods to adjust for the similarity among the siblings, and they can be clustered by the mothers.

Thank you. We would be concerned about restricting to maternal sibships. We have added to limitations p.22: Paternity or paternal sibships are not recorded, as in many databases. We considered each pregnancy separately, as exposures and covariates (e.g. smoking, deprivation) often vary between pregnancies. 

We accounted for parity in the analysis. 

3. Are any of these medication prescribed for other indications? Were there mothers in the cohort that received similar medication or discontinued these medications (but not for asthma)? This may have led to a misclassification of the studied exposure. If so, it would be interesting to study differences between the defined study groups and this group of women, in terms of the pregnancy outcomes etc. This could stress out whether the association was more likely with the medication/its` discontinuation, or the asthma itself. It would also be interesting to exclude such a group from the "unexposed" group in the study, and see how the results are affected.

I’m afraid these medicines are only indicated for asthma. In the notes to Table 1 we indicate: 

The only indications for these medicines is ‘conditions associated with reversible airways obstruction’ [28].

And p.23: … these medicines have no other indications in the UK. [28]

4. If the data is available, it would be interesting to model and compare women based on dosage of medication, or to compare women with no asthma/medication (reference group), and the following groups: women with untreated asthma, women discontinuing the medication, women receiving one medication, women receiving two medications, more than two medications. A dose response effect, if found, could strengthen the findings.

We agree. In the notes to Table 1, we have added to our text: We were unable to extract information on doses, so did not identify doses of oral medicines or step 4 [high dose ICS]: other inhalers are only available in standard doses. The only indications for these medicines is ‘conditions associated with reversible airways obstruction’ [28]. 

We were unable to extract information on doses, so did not identify step 4 [high dose ICS], or the doses of oral medicines: other inhaler are only available in standard doses. 

Although there is no information on doses, most of the inhalers in this study are standard doses. The women using just one medicine (SABA only or ICS only) are identified. 

We agree, we should address the dose-response issue, and have added to p.18. 

These data might support a dose-response vasoconstriction hypothesis, only partly dependent on hypoxia: infants exposed to more than one medicine, likely associated with more severe asthma, were rather more vulnerable to SGA. 

Few minor issues:

-line 4 (introduction) missing the word : Approximately 9% of pregnant women…

Done 

-line 21- SGA has been defined in previous paragraph.

Thank you, done

-line 174, it is unclear what the comparison group was. The unexposed? Pregnancies with ongoing asthma medication?

Thank you. We have clarified: , when compared with the whole population 

- line 294- missing a ].

Thank you, done

-line 318- unclear sentence, please rephrase.

Thank you, we have removed the sentence.

-changes in font size and style throughout the manuscript, as well as spacing between lines and paragraphs.

Thank you, done

Reviewer #4: This population-based study utilised maternal primary care data linked with data from >100,000 infants in Wales. The use and discontinuation of asthma medications for asthma in pregnancy was examined as the exposure, and adverse pregnancy outcomes and breastfeeding rates were examined.

This study adds additional information to our knowledge of the effects of asthma medication use on pregnancy outcomes, which is important given that there are very few RCTs on asthma medication use in pregnancy. Previous studies have not addressed the association of asthma or medication use for asthma and breastfeeding, an important novel aspect of this paper. 

Thank you.

However, breastfeeding information, while novel, was limited by a much smaller sample size (late introduction of data collection) and the available data was only able to be classified as any breastfeeding vs no breastfeeding.

We agree, and state this p.7. 

Breastfeeding in this database is defined as ‘any breastfeeding’ (exclusive or combined, predominantly or partially, with other feeds, including formula milk) as routinely recorded by health visitors at birth and 6-8 weeks and entered into NCCHD. [33]

The authors report in the introduction that they were unable to locate reports on the prevalence of breastfeeding amongst women with asthma. A recent publication in Pediatric Pulmonology (Harvey SM et al: Maternal asthma, breastfeeding and respiratory outcomes in the first year of life) contains this data. There are likely to be a few other papers from birth cohorts of infants at risk of asthma due to maternal asthma/atopy which may contain this information also. It would be helpful if the authors outlined in more detail the gap in this literature, including the recent paper from Harvey et al.

thank you. We have clarified P.4

We were unable to locate reports of the impact of prescription medicines on the prevalence of breastfeeding amongst women with asthma. 

We have added this useful reference at the point where we compared breastfeeding rates to current prevalence p.20.

Contemporary recruited cohorts elsewhere report higher breastfeeding rates amongst women with asthma, for example, 536/605, 88% at birth, 230/605 38% at 6 months (Harvey et al 2020).

Since the paper addresses the discontinuation of asthma medicine use in pregnancy, it would be good to see a paragraph in the introduction which outlines the extent of this clinical problem in pregnant women with asthma.

We agree this is important. We have now raised the issue in the introduction, and quoted prevalences. However, as we are advised to limit the introduction to 500 words, we have expanded in the discussion. 

Up to 50% pregnant women discontinue asthma medicines, often without professional advice, frequently worsening asthma and outcomes [1,9,14]. P.3

The associations between unmedicated asthma and premature birth support the consensus that uncontrolled asthma increases the risk of adverse perinatal outcomes subsequent to premature birth [3], but risks were not confined to severe asthma [5, Belanger et al 2010]. Rather, we highlight the importance of identifying the 28% of women with asthma who completely discontinue medicines, often ICS (Robinj et al 2019), as being at increased risk of premature birth and exclusive formula feeding; these women were apparently without symptoms sufficiently severe to warrant prescriptions. P.20

Belanger K, Hellenbrand ME, Holford TR, Bracken M. Effect of pregnancy on maternal asthma symptoms and medication use. Obstet Gynecol. 2010;115(3):559‐567. doi:10.1097/AOG.0b013e3181d06945

Robijn AL, Jensen ME, McLaughlin K, Gibson PG, Murphy VE. Inhaled corticosteroid use during pregnancy among women with asthma: A systematic review and meta-analysis. Clin Exp Allergy. 2019;49(11):1403‐1417. doi:10.1111/cea.13474

The authors state that “prescriptions reflect physicians’ assessment of asthma severity” (page 11, line 67). However, I am unsure if this assumption can be made for a pregnant population. Changes in asthma medication use specific to pregnancy would suggest that some physicians change prescribing behaviour in response to pregnancy itself. This needs to be addressed (as suggested above) in the introduction.

UK guidelines (BTS 2019) do not recommend changes in asthma medicines during pregnancy. UK GPs generally follow these guidelines, and compliance is monitored and remunerated by QOF (Quality Outcomes Framework) payments. We have added, p.5

… as prescribers adhere to guidelines [5] that direct monitoring and prescribing. 

Were LABA prescribed along with ICS, or as monotherapy? It wasn’t clear from table 1.

LABA are added to ICS. We have clarified in notes to Table 1.

There was a high proportion of women who received SABA only (45.4%). Do the authors have any comments on this in the context of recent changes to the GINA guidelines (people with mild asthma should not receive SABA alone)?

Thank you. This publication post dates our submission to Plos. We shall, of course, include such an important new reference. As indicated, our data on the adverse effects of SABA support the recommendations on p. 57 that SABA-only be no longer recommended at any stage p.17.

Preterm birth appeared more likely where only SABA were prescribed, but fell short of statistical significance, supporting recent guidelines no longer recommending ‘SABA only’ regimens for any asthma (GINA 2020 p.57).

Global Initiative for Asthma. Global Strategy for Asthma Management and Prevention. 2020, Available: https://ginasthma.org/wp-content/uploads/2020/04/GINA-2020-full-report_-final-_wms.pdf

The text on page 14, line 166-168 gives the numbers of women receiving ICS, LABA and LKA, and OCS – the proportions should also be added here.

We apologise, done

Were the women who discontinued medication the 3589 (28.3%) who received their last prescription before pregnancy (line 165)? It was not clear to me. Or was there another group who discontinued medications part way through pregnancy?

Thank you, the former is correct. We have clarified. 

and were deemed to have ‘discontinued’

The authors report that discontinuation of asthma medication before pregnancy was associated with preterm birth, but not with SGA. What is the proposed mechanism for these differing effects?

We think preterm birth is triggered by hypoxia, whereas SGA may be vasoconstriction, which is only partly determined by hypoxia. We discuss, p. 18 & 19

These [SGA] data might support a dose-response vasoconstriction hypothesis, only partly dependent on hypoxia: infants exposed to more than one medicine, likely associated with more severe asthma, were rather more vulnerable to SGA. 

The strongest predictors of premature birth were medicines discontinuation and OCS, supporting the hypothesis of a hypoxic trigger. The apparent protection against prematurity afforded by ICS suggests that ICS may have been controlling inflammation and asthma, thereby preventing hypoxia, and consequent CRH release, despite the potential to mimic endogenous corticosteroids. 

Breastfeeding prevalence at 6-8 weeks was lower in women with asthma medication before, during pregnancy, or discontinued in pregnancy. No further details were supplied in the text. This could be added, as table 6 only indicates a significant effect of asthma and unmedicated asthma in the adjusted odds ratios.

Thank you. we have added:

Prescriptions did not affect breastfeeding rates.

As the findings of this study focus on discontinuation of asthma medication, much more background information is needed on this issue, and more specifics on this are required in the methods/ results section.

Thank you. We have added information on prevalence of discontinuation, above. Discontinuation is defined in Table 1. Its impact is reported in tables 2-6, and discussed in the text. 

Some women voluntarily suspend medication during pregnancy, opting to tolerate symptoms rather than risk harming their unborn children. [58] Any uncontrolled asthma may cause chronic or intermittent hypoxia and inflammation, which adversely affect both mother and foetus, [56, 57] and may impair foetal development, causing intrauterine growth restriction, foetal distress, pre-term birth and, occasionally, death: stillbirths were more prevalent amongst women discontinuing asthma prescriptions [0.70%] when compared with UK rates 2000-2010 [0.51-0.58%] [59], and women never prescribed asthma medicines [0.38%], [S1 file, Table A1]. 

Non-adherence to asthma therapies exacerbates symptoms.[5,60] Absence of prescriptions for 9 months indicates withdrawal of therapy, as it is unlikely that stockpiled inhalers would be sufficient over this length of time. We do not know why women discontinued prescriptions or whether they experienced respiratory symptoms or they or their healthcare professionals attributed dyspnoea to pregnancy [61] or they remained well themselves, while their foetuses suffered episodes of hypoxia. Hypoxia stimulates release of placental corticotrophin-releasing hormone [CRH] and inflammatory cytokines. Placental CRH increases maternal and foetal cortisol production, which augments placental CRH release via positive feedback mechanisms, amplifying signals that may culminate in premature birth in ~5% infants. [52, 62] The strongest predictors of premature birth were medicines discontinuation and OCS, supporting the hypothesis of a hypoxic trigger

The discussion lacks critical review of the existing literature and is difficult to follow at times.

Thank you. We have revised and rechecked the literature. There are few papers reporting on discontinuation of asthma medicines in pregnancy, which cannot be conflated with non-adherence (see limitations). We should be please to follow specific advice.

The section on breastfeeding (line 303), describes these results as “neutral”; however, this was not the case as outlined in the results.

 Thank you, we have rephrased: 

Assurances [5,28] on the safety of SABA, ICS, most LABA, and OCS at <40mg per day may have contributed to the absence of a negative effect of prescriptions on breastfeeding.

 

Medicines prescribed for asthma, discontinuation and perinatal outcomes, including breastfeeding: a population cohort analysis

Short title: 

Asthma medicines, discontinuation and perinatal outcomes, including breastfeeding

Rebuttal letter PONE-D-19-17105 - [EMID:d126f82151db09f4]

30.01.20

Dear Professor Hashimoto,

Thank you for the return of our manuscript. We are very grateful for the reviewers’ support and suggestions. We have done our best to comply, as tabulated below. 

The sister paper, prepared from the same data set [1] has been published by Plos One, and has received some media coverage (below). The setting and flow diagram are inevitably the same for the two papers. This might present copyright issues if submitting to other journals. Accordingly, we are resubmitting to Plos One.

To our knowledge, we are the first to address the crucial question of medication discontinuation in pregnancy, and its impact on preterm birth, small for gestational age (SGA), and breastfeeding at 6-8 weeks. 

We feel that clinicians need to be aware of the importance of targeting women of childbearing age for medication review and additional support to reduce the risks of adverse outcomes, including exclusive formula feeding. In particular, we highlight that discontinuation of asthma medication during pregnancy increases risks of preterm delivery. Publication in PlosOne would stimulate prescribers to take a proactive approach to medication review in preconception care, an important, but neglected, component of the WHO 3rd Global Patient Safety Challenge on Medication Safety to reduce medicines-related harm by 50% by 2022 [2].

We hope that readers will concur with reviewer 2: As the asthma is a disease of many different phenotypes, severity and treatments, there is still need for more investigation such as this article.

The work has not been submitted elsewhere. 

Thank you for your time spent on this paper. 

Yours,

Sue Jordan, on behalf of all authors

[1] Jordan S, Davies GI, Thayer DS, Tucker D, Humphreys I (2019) Antidepressant prescriptions, discontinuation, depression and perinatal outcomes, including breastfeeding: A population cohort analysis. PLOS ONE 14(11): e0225133. https://doi.org/10.1371/journal.pone.0225133

Targeted News Service highlights a new study led by Swansea University’s Professor Sue Jordan on the impact of depression on pregnant women and new mothers (no link). Featured by EurekAlert. Also reported by News-Medical.net (Australia), 7thSpace (US), Doctor’s Guide (Canada).

[2] WHO 2017 Medication without harm. WHO, Geneva http://apps.who.int/iris/bitstream/10665/255263/1/WHO-HIS-SDS-2017.6-eng.pdf?ua=1&ua=1

Changes Made

Reviewer #1: 

The study covers a useful research question and the data source is appropriate for answering the key questions. From my perspective, however, the study would require substantial changes to be useful for the intended audience. Thank you.

These are my key observations: 

1) The objectives seem to be a mix of showing non causal associations and showing causal relationships. I think identifying associations is more appropriate without using more rigorous methods to account for confounding (e.g. propensity score methods). From this perspective, the current regression models would be what some people call “explanatory” models – i.e. to show various relationships – which is still useful. Some sections do, appropriately, focus on associations. Thank you, we agree that causality cannot be determined from retrospective cohorts. We have clarified our aims and methods:

To identify women likely to benefit from additional care, we... p.4

There was no randomisation, and, therefore, we explored association, rather than causality. P.4

The methods, though, describe control for confounding which would imply the goal is causal inference – i.e. to show the extent to which these medications cause the adverse pregnancy outcomes described. I would expect that all of the exclusion criteria would tend to limit generalizability, but I don’t see that they address the key confounding issues. I also don’t see that the regression models get at the key problems. When, for example, you want to know the causal association between SABA use and preterm birth, the article raises the issue of controlling for “confounding by indication.” In this case, SABAs could be used in patients with more severe asthma or with poorly controlled asthma, and therefore it is difficult to distinguish adverse effects of the drug from those of the underlying disease. 

To address confounding by indication you would need information on asthma severity (or other underlying causes of adverse outcomes) and use appropriate methods to address. The better alternative, from my perspective, is to simply treat this as descriptive data and not attempt to address confounding. You can still conduct multivariable modeling in the framework of a descriptive or explanatory regression model. You can also develop a “predictive” model but this would require a different strategy. Thank you. We now offer a definition of confounding from a standard medical text, based on an earlier Lancet series. Confounding is a blurring of effects when a variable is associated with the exposure and affects the outcome (Schulz & Grimes 2006 p.34). p.7

Restricting the dataset by excluding pregnancies exposed to AEDs, insulin, coumarins, heavy drinking or substance misuse and multiple pregnancies increases internal validity at the cost of generalisation to these groups (Schulz & Grimes 2006). P.17

Information on severity of asthma was inferred from prescriptions. As referenced, in the UK, prescriptions follow the BTS and BNF guidelines (references [5, 29]). P.5 and Table 1. Prescribers adhere to these guidelines. For example, oral corticosteroids are only prescribed for severe asthma. 

2) I would have expected the study to be conducted among women with asthma. Including all pregnant women made the interpretation more difficult. This is a population-based cohort. As healthcare databases become more available, such designs are increasingly common (Sterne et al 2016). We have added a sentence relating to their advantages p.4:

A population-based cohort was built from prospectively collected routine NHS data and analysed retrospectively. Such cohorts have minimal selection and attrition bias and no recall bias (Thygesen and Ersboll 2014). We explored association, rather than causality, as there was no randomisation.

3) I found several sections of the manuscript difficult to follow. In some cases, it was because multiple ideas were combined in the same sentence. Here is an example: “Breastfeeding [any, even if supplemented by formula feeding] at birth and 6- 8 weeks is routinely recorded by health visitors, and this information is transferred to NCCHD; data collection is more complete in some Health Board regions than others. [33] 

Breastfeeding at birth is considered an indirect measure, which may represent intention rather than practice, and is downgraded as evidence of successful breastfeeding. [34]” This section combined information on how data is collected (which I think fits better in the database description) and information on interpretation (does BF at birth correlate with later BF?), but doesn’t actually state how BF was defined for the analysis. I had trouble understanding what “indirect measure” meant (Indirect measure of longer term BF?) and what “is downgraded” meant (downgraded by other researchers?). And after all that, I still wasn’t clear on how BF was defined for the analysis, which was the main point of this section. 

As another example, the phrase “% without unknown outcome” was confusing. We have clarified, and added a reference p.6. Breastfeeding in this database is defined as ‘any breastfeeding’ (exclusive, or combined, predominantly or partially, with other feeds, including formula milk) as recorded by health visitors and entered into NCCHD (NHS Wales 2017). 

We have reworded to clarify, thank you: Breastfeeding at birth may represent intention to breastfeed, rather than actuality: a high proportion of dyads breastfeeding at birth have discontinued a week later. Therefore, this measure is downgraded as evidence of successful breastfeeding. [34] p.7

% without ‘unknown outcome’ is analogous to ‘missing or outcome unknown’. This entire table has been removed, as it is now available as supplementary material in the paper using the same dataset, but exploring antidepressants (Jordan et al 2019).

4) Starting the results and tables (e.g. Table 2) with an assessment of women excluded from your analysis was confusing. (Why focus the reader on these results if these women were not even eligible for the analysis?) I could see this assessment as a sensitivity analysis after presenting main results. This table is available in supplementary material in association with the co-submitted paper from the same data set. Therefore, it will not be submitted with this paper.

We believe this table contains very important information on the risks associated with other prescriptions, and will be of interest to those planning research in the area.

5) The results were not calculated and reported in a standard and useful way. If I look at the first 2 rows of Table 2, for example, a more useful way to present the results is:

a. Among those exposed to insulin, 213/688 (35.6%) had preterm delivery. (688 = 231+457)

b. Among those not exposed, 7863/112,628 (7.5%) had preterm delivery

c. The risk ratio (or relative risk) is then 4.48

d. The odds ratio (6.73) is technically correct but it is more difficult to interpret and could be easily interpreted as overstating the strength of the relationship. The regression models can match by using a model that reports a risk ratio. Modified Poisson is one that is easy to conduct: https://www.ncbi.nlm.nih.gov/pubmed/15033648 Log binomial can work as well although sometimes has convergence problems. This table has been removed from the paper. Readers are referred to its publication elsewhere.

We agree that odds and risk ratios can be used to calculate these differences, and readers are provided with sufficient information to make their own calculations. The numbers of subjects in table 2 are, in our opinions, too low for multivariate analyses. 

The tables report both the OR and the inverse of the OR which is confusing. In this case, the OR of 6.73 is the useful one for showing the relationship with increased risk of prematurity. The second OR (0.15) reversing the referent group is not useful. The tables should keep the referent group constant and identify the referent group (e.g. label cell as “ref”) and/or show an OR (or RR) of 1.0 for the referent group. The tables should also clearly delineate each association assessed. For example, the first 2 rows are one comparison (association between insulin and birth <37 weeks) and it would be helpful to distinguish that from subsequent comparisons. This table has been removed from the paper. Readers are referred to its publication elsewhere.

The additional data indicate to readers that data are completely reported. 

6) For each association, the comparison group should be stated clearly. In the abstract, for example, when you state that discontinuation is associated with preterm delivery, what is this in comparison to? (to those who continued all their treatments? To all other women including those without asthma or those who weren’t treated at all?) We are sorry if this is unclear. We state, p.8: For each analysis we compared the exposed with the unexposed population.

We have now added to the notes to table 2 and to the abstract: Odds ratios for adverse pregnancy outcomes were calculated for the exposed versus the unexposed population.

7) The data set seems quite old. Treatment guidelines and treatment options (such as biologics) have changed since this time. More recent data would be more relevant, if available. We agree. Were funding available, we would extend the study. However, this is the first report of the predictors of infant feeding and one of the most detailed reports available on medicine discontinuation in the ambulant population. We have used our data to identify how care can be improved, and hope that readers will consider this an important practice point. 

We have added to the ‘limitations’ p.18: Our data predate the primary care prescription of monoclonal antibodies for severe eosinophilic asthma persisting despite oral corticosteroid therapy. Manufacturers advise avoiding these medicines in pregnancy [29]; therefore, population databases may be unable to describe their safety in pregnancy for some time. 

Reviewer #2: 

The authors have found an association of adverse perinatal outcomes in primary care setting, in Brittan, Wales. It is an area with the greatest prevalence of asthma in Europe. As asthma is the most common chronic disease among pregnant women, we should be very careful on drawing conclusions. As the asthma is a disease of many different phenotypes, severity and treatments, there is still need for more investigation such as this article. Thank you. We agree, we cannot draw conclusions regarding aetiology, but we hope to suggest practice modifications. 

The authors have shown for the first time that the discontinuation of the medication of active asthma during pregnancy increases risk of preterm birth. There have been articles previously showing that unmedicated asthma has similar effects. The oral glucocorticoids are not analyzed in many of the articles evaluating asthma. The risks in the different articles are not seen similar in relation to oral glucocorticoids. The authors have shown for the first time that maternal asthma carries a risk of lower breast feeding rate for the first time. Thank you.

The authors present the literature in relation to asthma very well and their findings are in accordance to the previous literature. Thank you.

1) The terms hypoxia and asphyxia and their use should be controlled. As asthma may cause maternal hypoxia, the authors should be clear whether they mean maternal, placental, fetal, perinatal or neonatal hypoxia or asphyxia. This should be clear for the reader all times. Thank you, we have corrected. We have retained ‘asphyxia’ just once, on p.23, line 510, where complications of SGA are quoted: … range of conditions from neonatal asphyxia to mental health conditions in adulthood

2) The oral glucocorticoids are used for autoimmune diseases such as rheumatoid arthritis or inflammatory bowel diseases. These are associated with problems in pregnancies. Often people also have autoimmune tendency and suffer from several autoimmune diseases (asthma + RA+ IBD). It is also common for some patients to use short acting betasympathomimetics in pulmonary symptoms other than asthma. The authors have found association of worse prognosis associated with the oral glucocorticoid use. The authors have tried to make sure the use of oral corticosteroids is for asthma by necessitating the use of some other asthma medication. Have the authors information about the prevalence of the most common autoimmune diseases? If not, the authors should somehow comment the risk of confounding. We agree, this is an important point. Only 5 women with type 1 diabetes were prescribed oral corticosteroids (S1 file, table A1). Only 519 pregnancies were exposed to oral corticosteroids. Although we excluded the commonest co-morbidities, where pregnancy risks are known to be higher than for asthma, we acknowledge that other, rarer, co-morbidity was not excluded, and have added to our ‘limitations’ section: 

Rarer co-morbidities were not excluded. Co-morbid rheumatoid arthritis and inflammatory bowel disease were more likely amongst women prescribed oral corticosteroids, who would have received specialist care [5}. 

We agree that beta2 adrenoceptor agonists are indicated for all conditions associated with reversible airways obstruction. In this age-group, asthma is the most common. We have added to notes to Table 1:

SABA and LABA are indicated for all conditions associated with reversible airways obstruction [29]. 

And to our discussion, final paragraph:

Prescription patterns, regardless of diagnoses, particularly discontinuation in pregnancy, offer primary care nurses, midwives and nurse specialists a convenient marker to identify pregnancies at increased risk, analogous to an ‘intention to treat’ analysis: pragmatic but of unproven biological certainty. 

4) Authors state in introduction without a reference that asthma medications cross placenta and enter breastmilk. Is this true for all the asthma medications and could the sentence just be left out? As the article is not about congenital anomalies, is it necessary to add articles about them in relation to asthma? We apologise, and have added references. We suggest this short sentence supports the biological plausibility of our findings.

We have included the reference to congenital anomalies to alert future systematic reviewers, that this data set has been used for this outcome. It is important that future reviewers avoid all possible double-counting of events. 

5) Did the authors have information on just prescription or also on purchases and dispension? Please state clearly. We apologise if this is not clear. We have clarified:

We have no data on dispensing or hospital or private prescribing or internet purchasing. It is unlikely that anyone hospitalized with asthma or consulting privately would not be receiving prescriptions from primary care practitioners. Abolition of prescription charges in Wales in 2007 reduces the likelihood of private prescribing and purchasing.

6) This setting is primary care setting. The amounts of mediccations used differ a bit from the other studies. The secondary and tertiary care prescribtions on asthma medications are not included? Does this explain absence of some medication groups? Most of the asthma is managed in primary care, which make primary care settings more useful. We hope the sentence above clarifies. 

We agree, that our important findings relate to primary care, and the ‘less ill’ women. We think that even a 1% increase in preterm birth rate in these women is important, and might be avoided.

Jordan S, Davies GI, Thayer DS, Tucker D, Humphreys I (2019) Antidepressant prescriptions, discontinuation, depression and perinatal outcomes, including breastfeeding: A population cohort analysis. PLOS ONE 14(11): e0225133. https://doi.org/10.1371/journal.pone.0225133

Schulz KF and Grimes DA 2006 The Lancet Handbook of Essential concepts in Clinical Research. Elsevier, Edinburgh.

Sterne Jonathan AC, Hernán Miguel A, Reeves Barnaby C, Savović Jelena, Berkman Nancy D, Viswanathan Meera et al. ROBINS-I: a tool for assessing risk of bias in non-randomised studies of interventions BMJ 2016; 355 :i4919

NHS Wales Data Dictionary 2017 http://www.datadictionary.wales.nhs.uk/#!WordDocuments/infantfeedingdatarequirements.htm

Thygesen, L.C. & Ersbøll, A.K. When the entire population is the sample: strengths and limitations in register-based epidemiology Eur J Epidemiol (2014) 29: 551. https://doi.org/10.1007/s10654-013-9873-0

notes

Search: asthma pregnancy brestfeeding Filters: Review, Systematic Reviews, English Sort by: Most Recent

---

## [Editor Report · Decision Letter 1]

21 Jul 2020

PONE-D-20-02812R1

Medicines prescribed for asthma, discontinuation and perinatal outcomes, including breastfeeding: a population cohort analysis

PLOS ONE

Dear Dr. Jordan,

Thank you for submitting your manuscript to PLOS ONE. After careful consideration, we feel that it has merit but does not fully meet PLOS ONE’s publication criteria as it currently stands. Therefore, we invite you to submit a revised version of the manuscript that addresses the points raised during the review process.

Please see comments below.

We look forward to receiving your revised manuscript.

Kind regards,

Abigail Fraser

Academic Editor

PLOS ONE

Additional Editor Comments (if provided):

Thank you for your efforts to revise the manuscript. That said, several points still need to be addressed

1. Adjusting for age – whilst age might not be modifiable, it may be an important factors to identify high risk groups. Authors should explore options of categorising age into bands, entering different variables in separate models (to avoid collinearity), etc.

2. It is not enough to note a limitation without spelling out to readers how this might affect results (e.g. result in underestimation of overestimation).

3. NNH should be removed from the manuscript.

4. The term statistical significance should be defined (ie the threshold specified) or even better, removed entirely.

---

## [Author Response · Author response to Decision Letter 1]

27 Aug 2020

Response to editor

 Response 

1. Adjusting for age – whilst age might not be modifiable, it may be an important factors to identify high risk groups. Authors should explore options of categorising age into bands, entering different variables in separate models (to avoid collinearity), etc.

 Age in bands was entered into the logistic regression models. It was a significant predictor of breastfeeding, but not other outcomes. Results have been adjusted. 

2. It is not enough to note a limitation without spelling out to readers how this might affect results (e.g. result in underestimation of overestimation).

 In some instances, this was straightforward; in others, there is little information or the estimation was complicated by the association between medicines discontinuation and harms. New text is red.

Restricting the dataset by excluding pregnancies exposed to AEDs, insulin, coumarins, heavy drinking or substance misuse and multiple pregnancies increased internal validity without introducing bias, at the cost of generalisation to these groups. [37, 83] The higher prevalence of these exposures amongst exposed women (S1 file, Table A1) would have caused overestimation of harms attributable to asthma medicines.

Maternal stress, short and long-term, increases the risk of pre-term birth. [66, 79] Substance misuse is notoriously difficult to capture, and may be under-reported; however, it is often persistent. We excluded the most vulnerable women [39-42]. The proportion of missing data, [35.5% overall, 29.3% in women with asthma], precluded inclusion of BMI in regression analyses [S1 file, Table A1]. Any increased prevalence of these predictors of adverse outcomes amongst exposed women would lead to over-estimation of associations. Paternity and paternal sibships are not recorded, as in many databases, but impact on perinatal outcomes are believed to be modest (Oldereid et al 2018). We considered each pregnancy separately, as exposures and covariates (e.g. smoking, deprivation) often vary between pregnancies. We acknowledge the genetic determinants of perinatal outcomes (Liu et al 2019), but, without information on their association with asthma, are unable to predict how accounting for this would affect this study.

No data were available on miscarriage and induced terminations. Confining analyses to otherwise healthy live-born infants introduces a survival / survivorship bias, leaving some harms unreported, and under-estimating overall harm [85]. This was confirmed by the slightly higher rates of stillbirth (S1 file, Table A1). Congenital anomalies and stillbirths are associated with SGA and preterm birth, [86], and the higher prevalence of congenital anomalies amongst women exposed to asthma medicines in this cohort is known; [19] therefore, affected infants were excluded. 

Non-adherence would dilute the impact of medicines on outcomes; however, since discontinuation was associated with harm, unreported discontinuation in those receiving prescriptions might lead to an over-estimation of harms associated with prescriptions. 

3. NNH should be removed from the manuscript.

 This has been done.

4. The term statistical significance should be defined (ie the threshold specified) or even better, removed entirely. We have added under ‘statistical analysis’: “Odds ratios, unadjusted and adjusted are reported together with 95% confidence / compatibility intervals.” 

We agree, the term ‘statistical significance’ is becoming outmoded. We have substituted “interval estimates included no difference”, in accordance with ASA advice [1]. 

Ronald L. Wasserstein, Allen L. Schirm & Nicole A. Lazar (2019) Moving to a World Beyond “p < 0.05”, The American Statistician, 73:sup1, 1-19, DOI: 10.1080/00031305.2019.1583913

Abigail Fraser

Academic Editor

PLOS ONE

Dear Dr. Fraser,

We are returning our paper, having addressed the comments of the 4 reviewers and editor, below. As the reviewers indicate, this is the first report of the impact of asthma medicines on breastfeeding, and we have important data on medicines discontinuation. We are particularly grateful where reviewers have suggested particular references. At times, we have been unable to identify literature recommended. We should, therefore, be very pleased were the reviewers to find the time to indicate which articles they feel should be referenced.

As we indicated, we were invited to resubmit this paper last year, when we addressed the concerns of the 2 earlier referees. Our responses to these are appended below. A paper on antidepressants from this cohort was submitted as a sister paper to this, and was published last year in Plos One. The overlap is in the cohort, and we have added this information to the paper. 

We should like to thank you and the reviewers for their time spent on our work. Please let us know if further changes are needed.

Yours, 

Sue Jordan

Yr Athro Professor 

Coleg y Gwyddorau Dynol a Iechyd College of Human and Health Sciences

Prifysgol Abertawe Swansea University

Abertawe SA2 8PP Swansea SA2 8PP

01792 518541 01792 518541 

Cynghorydd Cymuned Gwaun Cae Gurwen Community Councillor, Gwaun Cae Gurwen

Llywydd (ar y cyd) Cangen Undeb y Brifysgol a'r Coleg Branch President (joint) UCU 

project website: http://www.swansea.ac.uk/adre/

Editorial Office

We noted in your submission details that a portion of your manuscript may have been presented or published elsewhere:

'Antidepressant prescriptions, discontinuation, depression and perinatal outcomes, including breastfeeding: A population cohort analysis (2019) Sue Jordan, Gareth I. Davies, Daniel S. Thayer David Tucker, Ioan Humphreys '

Please clarify whether this publication was peer-reviewed and formally published. If this work was previously peer-reviewed and published, in the cover letter please provide the reason that this work does not constitute dual publication and should be included in the current manuscript.

We noted this publication in Plos One in our covering letter, below, and our original submission to Plos One in 2019. The paper was peer-reviewed. We explain that the 2 papers emanate from the same cohort. Therefore, the study flow diagram, and the setting and participants are the same

Jordan S, Davies GI, Thayer DS, Tucker D, Humphreys I (2019) Antidepressant prescriptions, discontinuation, depression and perinatal outcomes, including breastfeeding: A population cohort analysis. PLOS ONE 14(11): e0225133. https://doi.org/10.1371/journal.pone.0225133

We have added to p.5:

The impact of antidepressants on breastfeeding, premature birth and SGA in this cohort has been reported [28, Jordan et al 2019].

Editor

Please pay particular attention to concerns shared by the reviewers: the potential selection bias introduced by excluding stillbirths and severe congenital malformations; 

We have added to limitations p.22

No data were available on miscarriage and induced terminations. Confining analyses to otherwise healthy live-born infants introduces a survival / survivorship bias, leaving some harms unreported (Schnitzer & Blais 2018). This was confirmed by the slightly higher rates of stillbirth (S1 file, Table A1). Congenital anomalies and stillbirths are associated with SGA and preterm birth [Miquel-Verges et al 2015]; therefore, affected infants were excluded. 

the conflations of disease severity and medication type; we added on p.18

The associations between LABA and LKA and SGA <3rd centile may be confounded by asthma severity. These data might support a dose-response vasoconstriction hypothesis, only partly dependent on hypoxia: infants exposed to more than one medicine, likely associated with more severe asthma, were rather more vulnerable to SGA. 

And on p.24

In evaluating the impact of medicines using population databases, it is rarely possible to avoid residual confounding by indication and severity of indicationwithout randomisation [85,86]

the need to clarify why asthma medication discontinuation in pregnancy might be problematic or important; 

we have added to the introduction:

Up to 50% pregnant women discontinue asthma medicines, often without professional advice, frequently worsening asthma and outcomes [1,9,14]. 

And discuss on p.19 

The strongest predictors of premature birth were medicines discontinuation and OCS, supporting the hypothesis of a hypoxic trigger. Et seq

clarifying the comparisons/reference groups.

We have written, p.8: For each analysis we compared the exposed with the unexposed population. This allows different medicines to be compared. 

Please also note reviewer 1's concerns about a causal interpretation. If you, the authors are not committed to a causal interpretation, then a different justification for the study will need to be provided. 

Our aim is to identify how best to target support, p.4:

To identify women likely to benefit from additional care, and target support, we aimed to explore any associations between prescriptions for asthma medicines during pregnancy, or their discontinuation, and the range of perinatal outcomes that matter to women [prematurity, SGA, and, for the first time, breastfeeding at 6-8 weeks, full or partial]. 

We feel that we have achieved this objective:

Prescription patterns, regardless of diagnoses, particularly discontinuation in pregnancy, offer primary care practitioners a convenient marker to identify pregnancies at increased risk, analogous to ‘intention to treat’ analyses: pragmatic but of unproven biological certainty. P.21

Taken with lower breastfeeding rates, these findings identify unmet needs amongst women discontinuing prescriptions, and militate against medication reduction in pregnancy [87]. Reduced risks with ICS suggest that increased monitoring, targeted support and active asthma management [3, 87, GINA 2020] are needed before, during and after pregnancy. Programming primary care electronic records to alert professionals to contact women who leave >4 months between prescriptions for asthma medicines might make this feasible. P.24

If this could be published, we can present the evidence to the all-Wales committees to ask for this to be done. 

Reviewer #1: The authors used a secure anonymous information linkage from wales to examine whether women who use asthma medication have an increased risk of adverse pregnancy outcomes. They report an increased risk of gestational age <32 weeks and small-for-gestational age (SGA) among women who use asthma medications during pregnancy, in addition to a lower likelihood of breastfeeding. Women who discontinued their use of asthma medications during pregnancy were found to have a particularly increased risk of preterm birth and SGA.

1) The authors chose to exclude all stillbirths and all live births with malformations. These are other pregnancy outcomes that could be on the causal pathway for the relationship between use of asthma medications during pregnancy and the adverse pregnancy outcomes the authors are studying. It would be good if the authors could better explain their rational, particularly due to the fact that these exclusions could have introduced bias (specifically collider bias). 

Thank you. We have added to p.5:

Infants with congenital anomalies were excluded, as associations with asthma prescriptions are reported elsewhere. [19]

Reporting twice on the same cases jeopardises any future meta-analyses. 

And to limitations p.22

No data were available on miscarriage and induced terminations. Confining analyses to otherwise healthy live-born infants introduces a survival / survivorship bias, leaving some harms unreported (Schnitzer & Blais 2018). This was confirmed by the slightly higher rates of stillbirth (S1 file, Table A1). Congenital anomalies and stillbirths are associated with SGA and preterm birth [Miquel-Verges et al 2015]; therefore, affected infants were excluded. 

Schnitzer ME, Blais L. Methods for the assessment of selection bias in drug safety during pregnancy studies using electronic medical data. Pharmacol Res Perspect. 2018;6(5):e00426. Published 2018 Sep 21. doi:10.1002/prp2.426 https://www.ncbi.nlm.nih.gov/pmc/articles/PMC6149369/

Miquel-Verges F, Mosley BS, Block AS, Hobbs CA. A spectrum project: preterm birth and small-for-gestational age among infants with birth defects. J Perinatol. 2015 Mar;35[3]:198-203. doi: 10.1038/jp.2014.180. Epub 2014 Oct 2. PubMed PMID: 25275696.

2) Did any women contribute to the analysis with more than one pregnancy ? If so, how was this accounted for?

Thank you. We have added to limitations p.22: Information on paternity or paternal sibships is incomplete, as in many databases. We considered each pregnancy separately, as exposures and covariates (e.g. smoking, deprivation) often vary between pregnancies. 

We accounted for parity in the analysis. 

3) I do not understand why the authors chose to not account for maternal age. Their indirect claim that this additional adjustment would not influence the associations seems insufficient without presenting how the results changed with this additional adjustment.

Thank you. None of the other 5 reviewers have made this request. Age is not a modifiable risk factor. The association with deprivation would adversely affect the collinearity of the regression models, as explained, p.7 line 131 et seq:

Maternal age, a non-modifiable risk factor, is strongly associated with SES and parity in this cohort, and there were too few women aged >39 at birth to account for this potentially high risk group separately. [42] Therefore, in view of the low numbers in some outcomes, to reduce collinearity with SES and parity, age was not entered into the regression analyses.

 We therefore focussed on the modifiable risk factors, including deprivation. We have added this to our ‘limitations’ section. We acknowledge that not including data on age detracts from the regression models, but parsimony is important where there are low numbers of exposed cases. We have added, with supporting references:

We omitted age from the models to reduce collinearity and maintain parsimony, as numbers of exposed cases were low. Observation studies are limited by unmeasured confounding, and the impossibility of including all contextual variables (York 2012, Greenland et al 2016); we opted to retain deprivation as a key policy informant.

Greenland S, Daniel R, Pearce N. Outcome modelling strategies in epidemiology: traditional methods and basic alternatives. Int J Epidemiol. 2016;45(2):565‐575. doi:10.1093/ije/dyw040

 York R. Residualization is not the answer: Rethinking how to address multicollinearity. Soc Sci Res. 2012;41(6):1379‐1386. doi:10.1016/j.ssresearch.2012.05.014

4) On the one hand, the authors say clearly that what they are presenting are associations and not causal effects, but on the other hand they choose to present the number needed to harm (NHH). Based on their limited opportunity to account for potential confounding factors, and the high likelihood of unmeasured confounding, I strongly recommend that they take these estimations out of the paper. 

Thank you. We await the editors’ advice. These calculations of the absolute risk differences are recommended in some journals and are compulsory in others, as they offer greater clarity than odds ratios. NNH conveys absolute risks, which are important to women. It is presented in the sister paper in this journal (Jordan et al 2019). None of the 5 other reviewers have asked for removal of these calculations. We should prefer to allow readers to judge the utility of the parameter for themselves. We have added the rubric, below (p.8):

To report absolute risk differences, which are important to women, the number needed to harm [NNH] was calculated for statistically significant results.

The authors should also state that their lack of information on BMI is an important limitation of the study.

We agree. The paucity of BMI data was noted as a limitation p.22: The proportion of missing data, [35.5% overall, 29.3% in women with asthma], precluded inclusion of BMI in regression analyses [S1 file, Table A1]. 

5) Can the authors please confirm that they did not have any information available on other allergic diseases that would allow them to try to distinguish between allergic and non-allergic asthma?

Thank you, we had written (Lines 422 et seq)

Without data on medicines purchased without prescription, we could not explore the hypothesis that ‘asthma with hayfever’ is a stronger predictor of premature birth than non-atopic asthma [83]. 

We have now added: Similarly, paucity of testing for exhaled nitric oxide, which may not improve outcomes [14], made it impossible to ascribe eosinophilic or allergic aetiology to the diagnosis. 

6) It would be great if the authors could make it even more clear how the different medications might reflect the severity of the underlying asthma symptoms. How can the results across the different medications groups be interpreted as reflecting disease severity?

Thank you. We have added p.5: 

Prescriptions reflect physicians’ assessment of asthma severity [5, 28], since prescribers adhere to guidelines [5] that direct monitoring and prescribing.

The notes to table 1 detail the UK prescribing guidelines, which are referenced. 

7) I think that the authors need to be careful in drawing the conclusion that discontinuation of asthma medications during pregnancy is particularly harmful. They do not know whether these women had mild symptoms, nor indeed how well any of the women were regulated due to their lack of information on asthma symptoms. I therefore consider their second sentence in the conclusion to be an overstatement (“Identifying women discontinuing asthma medicines during pregnancy from prescription records and offering close monitoring and support for breastfeeding might address these problems.”). This sentence should be deleted and the discussion appropriately revised.

Thank you, we have redrafted: Women discontinuing medicines during pregnancy could be identified from prescription records. The impact of targeting close monitoring and breastfeeding support warrants exploration. 

8) Would it be possible to conduct a comparison of the risk of adverse pregnancy outcomes and the likelihood of breastfeeding across the groups of or combinations of asthma medications? The reference group is now those who do not use asthma medications. It could be interesting to compare the risk between women using different asthma medications directly, to see if any of the specific medications pose a particularly high risk.

Thank you. We have now clarified in the notes to Table 1 that some medicines (LABA, LKA, OCS) are not prescribed alone. We have distinguished where ICS and SABA were prescribed alone. We have written, p.8: For each analysis we compared the exposed with the unexposed population. This allows different medicines to be compared, as requested. 

9) Are the authors able to identify the subgroup of women first diagnosed with asthma during pregnancy? There is literature indicating that this group of women might have a particularly high risk of adverse pregnancy outcomes.

Thank you. Unfortunately, we did not identify women with an earlier diagnosis, because the database only extends to 2000, missing any childhood asthma. Women diagnosed with asthma for the first time during pregnancy are likely to have had earlier mild symptoms overlooked. We have added to limitations: 

The timeframe did not allow us to distinguish women receiving their first ever prescription during pregnancy. 

Minor comments

1) It is necessary that a bit more information about the databases is presented in the methods to allow readers to fully appreciate the study population.

We agree, but word limit is a problem, and there are several papers on this, references 25 & 26. I’m not sure as to the exact information required, so we’ve added the link to the SAIL FAQs https://saildatabank.com/faq/. We fully describe SAIL in our EUROmediCAT report, reference 27. We have added weblinks to data sources. 

Reviewer #2: Thank you for giving me the opportunity to review this article, which aimed to explore associations between medication (and discontinuation of) use for asthma in pregnancy and pregnancy outcomes

This population based cohort study demonstrated an association between medication use for asthma and preterm birth(<32 weeks), and SGA, whereas discontinuation of the medications was associated with preterm birth (both <37 weeks and 32 weeks ) but not SGA.

I believe the aim of the study is important and it has been conducted well and rigorously , 

Thank you

however I believe the reasons for non medication use, and discerning this from asthma disease severity are not well described in the article and therefore needs more clarity in the discussions and conclusions.

Thank you. We agree, but have no information as to why medicines were discontinued or if symptoms arose. We have added to limitations p.18:

We do not know why women discontinued prescriptions or whether they suffered respiratory symptoms or they or their healthcare professionals attributed dyspnoea to pregnancy [61] or they remained well themselves, while their foetuses may have suffered episodes of hypoxia. 

More detailed comments on the article;

Introduction

I believe the introduction is missing the essential concept that active poorly uncontrolled asthma is associated with preterm birth and SGA, and that medication use, especially multiple medication use is likely reflective of more poorly controlled disease. The impact of the actual medication itself on such outcomes is a separate influencing factor.

Thank you, we agree. We have moved from the discussion to introduction p.3:

uncontrolled asthma increases the risk of adverse perinatal outcomes, [3]

We continue, duly modified: we aimed to explore any associations between prescriptions for asthma medicines during pregnancy, or their discontinuation

under ‘exposure p.5 we acknowledge:

Prescriptions reflect physicians’ assessment of asthma severity [5, 28].

And give details in Table 1 and its footnotes. 

Page 4 line 37 – congenital anomalies are rare, was this a main aim of the study, and major congenital abnormalities were excluded?

We explain p.5: 

Infants with congenital anomalies were excluded, as associations with asthma prescriptions are reported elsewhere. [19]

Methods

Page 4 line 40 – suggest moving this sentence to the discussion

Thank you, done. 

Page 4 line 49- is there a bias in the 60% of practices that did not share the data?

Thank you, yes, there was a small difference, reported in reference 27. We have now added to p.4: 

Women in the included population were slightly less deprived and older than the rest of Wales [27].

Page 7 line 114 – this line should be removed

Done 

Page 7 – line 118 – why exclude stillbirth

Congenital anomalies in these cohorts were reported elsewhere (reference 19). We should have clarified this on p.5, and have now added.

Infants with congenital anomalies were excluded, as associations with asthma prescriptions are reported elsewhere. [19]

Table 2 Premature birth

corrected – thank you

Discussion

Overall discussion needs to be more concise

We have reviewed and reduced, but needed to address reviewers’ very valid points. We addressed each outcome, the issue of unmedicated asthma and the limitations of observational work. The latter have been expanded to address the thoughtful contributions of the 6 reviewers. 

Page 17 line 239 – I am unsure this statement is entirely correct, women may discontinue medication in pregnancy f they have minimal symptoms but they may also discontinue medication if they are concerned about the impact of the medications on the pregnancy, this in turn can cause uncontrolled asthma with worsening severity and potentially worse pregnancy outcomes in itself.

We agree. This sentence has been removed.

Page 17 – line 252, which can in turn cause worsening disease

Thank you, added.

Page 17 line 258 – see above statement for line 239

Thank you, this sentence has been deleted.

Page 18 – 265- how can you separate these effects from disease severity

We agree, there is a high risk of confounding by severity here: we only suggest further work to explore. We have added:

The associations between LABA and LKA and SGA <3rd centile may be confounded by the severity of asthma, but warrant further investigation.

Page 18 line 266 – this section regarding non-adherence needs to be concisely described in the introduction

Thank you. We have moved from the discussion to introduction p.3:

uncontrolled asthma increases the risk of adverse perinatal outcomes, [3] (…) and added

Up to 50% pregnant women discontinue asthma medicines, often without professional advice, frequently worsening asthma and outcomes [1,9,14]. 

Many journals ask authors to confine the introduction to 500 words, but do not restrict the discussion. Unfortunately, we have no information on adherence, and this is included with the other limitations.

Reviewer #3: This is a large population based study, on the association between exposure to asthma medication, its discontinuation, and pregnancy outcomes, including breastfeeding. This is a relatively prevalent exposure, and the question important for healthcare providers and women planning a pregnancy. The researchers were able to include data on breastfeeding at age 6-8 weeks, which is another public health concern, and to clarify whether it is affected by maternal medication status and possible knowledge on this issue.

This is an important subject, the manuscript is well written, and the Methods and Results are clear and seem appropriate for this question. 

Thank you

I do have a few suggestions, that need clarification or additions to the manuscript:

1. If I understand correctly, pregnancy terminations were excluded from the study, does this include stillbirths? If so, it is possible the worse outcomes were excluded from the study, and the live births included in this study are actually present a survival bias. Again, if this is the case, this possible survival bias, and it`s effect on the results needs to be addressed in the Discussion section.

Thank you. Yes, this is correct. The numbers of stillbirths are reported in S1 file, Table A1, and commented in the discussion p.18. We have added to limitations p.22:

No data were available on miscarriage and induced terminations. Confining analyses to otherwise healthy live-born infants introduces a survival / survivorship bias, leaving some harms unreported (Schnitzer & Blais 2018). This was confirmed by the slightly higher rates of stillbirth (S1 file, Table A1). Congenital anomalies and stillbirths are associated with SGA and preterm birth [Miquel-Verges et al 2015]; therefore, affected infants were excluded. 

Schnitzer ME, Blais L. Methods for the assessment of selection bias in drug safety during pregnancy studies using electronic medical data. Pharmacol Res Perspect. 2018;6(5):e00426. Published 2018 Sep 21. doi:10.1002/prp2.426 https://www.ncbi.nlm.nih.gov/pmc/articles/PMC6149369/

Miquel-Verges F, Mosley BS, Block AS, Hobbs CA. A spectrum project: preterm birth and small-for-gestational age among infants with birth defects. J Perinatol. 2015 Mar;35[3]:198-203. doi: 10.1038/jp.2014.180. Epub 2014 Oct 2. PubMed PMID: 25275696.

2. Were there siblings in the cohort? How were they addressed, as they are not independent one from another. There are statistical methods to adjust for the similarity among the siblings, and they can be clustered by the mothers.

Thank you. We would be concerned about restricting to maternal sibships. We have added to limitations p.22: Paternity or paternal sibships are not recorded, as in many databases. We considered each pregnancy separately, as exposures and covariates (e.g. smoking, deprivation) often vary between pregnancies. 

We accounted for parity in the analysis. 

3. Are any of these medication prescribed for other indications? Were there mothers in the cohort that received similar medication or discontinued these medications (but not for asthma)? This may have led to a misclassification of the studied exposure. If so, it would be interesting to study differences between the defined study groups and this group of women, in terms of the pregnancy outcomes etc. This could stress out whether the association was more likely with the medication/its` discontinuation, or the asthma itself. It would also be interesting to exclude such a group from the "unexposed" group in the study, and see how the results are affected.

I’m afraid these medicines are only indicated for asthma. In the notes to Table 1 we indicate: 

The only indications for these medicines is ‘conditions associated with reversible airways obstruction’ [28].

And p.23: … these medicines have no other indications in the UK. [28]

4. If the data is available, it would be interesting to model and compare women based on dosage of medication, or to compare women with no asthma/medication (reference group), and the following groups: women with untreated asthma, women discontinuing the medication, women receiving one medication, women receiving two medications, more than two medications. A dose response effect, if found, could strengthen the findings.

We agree. In the notes to Table 1, we have added to our text: We were unable to extract information on doses, so did not identify doses of oral medicines or step 4 [high dose ICS]: other inhalers are only available in standard doses. The only indications for these medicines is ‘conditions associated with reversible airways obstruction’ [28]. 

We were unable to extract information on doses, so did not identify step 4 [high dose ICS], or the doses of oral medicines: other inhaler are only available in standard doses. 

Although there is no information on doses, most of the inhalers in this study are standard doses. The women using just one medicine (SABA only or ICS only) are identified. 

We agree, we should address the dose-response issue, and have added to p.18. 

These data might support a dose-response vasoconstriction hypothesis, only partly dependent on hypoxia: infants exposed to more than one medicine, likely associated with more severe asthma, were rather more vulnerable to SGA. 

Few minor issues:

-line 4 (introduction) missing the word : Approximately 9% of pregnant women…

Done 

-line 21- SGA has been defined in previous paragraph.

Thank you, done

-line 174, it is unclear what the comparison group was. The unexposed? Pregnancies with ongoing asthma medication?

Thank you. We have clarified: , when compared with the whole population 

- line 294- missing a ].

Thank you, done

-line 318- unclear sentence, please rephrase.

Thank you, we have removed the sentence.

-changes in font size and style throughout the manuscript, as well as spacing between lines and paragraphs.

Thank you, done

Reviewer #4: This population-based study utilised maternal primary care data linked with data from >100,000 infants in Wales. The use and discontinuation of asthma medications for asthma in pregnancy was examined as the exposure, and adverse pregnancy outcomes and breastfeeding rates were examined.

This study adds additional information to our knowledge of the effects of asthma medication use on pregnancy outcomes, which is important given that there are very few RCTs on asthma medication use in pregnancy. Previous studies have not addressed the association of asthma or medication use for asthma and breastfeeding, an important novel aspect of this paper. 

Thank you.

However, breastfeeding information, while novel, was limited by a much smaller sample size (late introduction of data collection) and the available data was only able to be classified as any breastfeeding vs no breastfeeding.

We agree, and state this p.7. 

Breastfeeding in this database is defined as ‘any breastfeeding’ (exclusive or combined, predominantly or partially, with other feeds, including formula milk) as routinely recorded by health visitors at birth and 6-8 weeks and entered into NCCHD. [33]

The authors report in the introduction that they were unable to locate reports on the prevalence of breastfeeding amongst women with asthma. A recent publication in Pediatric Pulmonology (Harvey SM et al: Maternal asthma, breastfeeding and respiratory outcomes in the first year of life) contains this data. There are likely to be a few other papers from birth cohorts of infants at risk of asthma due to maternal asthma/atopy which may contain this information also. It would be helpful if the authors outlined in more detail the gap in this literature, including the recent paper from Harvey et al.

thank you. We have clarified P.4

We were unable to locate reports of the impact of prescription medicines on the prevalence of breastfeeding amongst women with asthma. 

We have added this useful reference at the point where we compared breastfeeding rates to current prevalence p.20.

Contemporary recruited cohorts elsewhere report higher breastfeeding rates amongst women with asthma, for example, 536/605, 88% at birth, 230/605 38% at 6 months (Harvey et al 2020).

Since the paper addresses the discontinuation of asthma medicine use in pregnancy, it would be good to see a paragraph in the introduction which outlines the extent of this clinical problem in pregnant women with asthma.

We agree this is important. We have now raised the issue in the introduction, and quoted prevalences. However, as we are advised to limit the introduction to 500 words, we have expanded in the discussion. 

Up to 50% pregnant women discontinue asthma medicines, often without professional advice, frequently worsening asthma and outcomes [1,9,14]. P.3

The associations between unmedicated asthma and premature birth support the consensus that uncontrolled asthma increases the risk of adverse perinatal outcomes subsequent to premature birth [3], but risks were not confined to severe asthma [5, Belanger et al 2010]. Rather, we highlight the importance of identifying the 28% of women with asthma who completely discontinue medicines, often ICS (Robinj et al 2019), as being at increased risk of premature birth and exclusive formula feeding; these women were apparently without symptoms sufficiently severe to warrant prescriptions. P.20

Belanger K, Hellenbrand ME, Holford TR, Bracken M. Effect of pregnancy on maternal asthma symptoms and medication use. Obstet Gynecol. 2010;115(3):559‐567. doi:10.1097/AOG.0b013e3181d06945

Robijn AL, Jensen ME, McLaughlin K, Gibson PG, Murphy VE. Inhaled corticosteroid use during pregnancy among women with asthma: A systematic review and meta-analysis. Clin Exp Allergy. 2019;49(11):1403‐1417. doi:10.1111/cea.13474

The authors state that “prescriptions reflect physicians’ assessment of asthma severity” (page 11, line 67). However, I am unsure if this assumption can be made for a pregnant population. Changes in asthma medication use specific to pregnancy would suggest that some physicians change prescribing behaviour in response to pregnancy itself. This needs to be addressed (as suggested above) in the introduction.

UK guidelines (BTS 2019) do not recommend changes in asthma medicines during pregnancy. UK GPs generally follow these guidelines, and compliance is monitored and remunerated by QOF (Quality Outcomes Framework) payments. We have added, p.5

… as prescribers adhere to guidelines [5] that direct monitoring and prescribing. 

Were LABA prescribed along with ICS, or as monotherapy? It wasn’t clear from table 1.

LABA are added to ICS. We have clarified in notes to Table 1.

There was a high proportion of women who received SABA only (45.4%). Do the authors have any comments on this in the context of recent changes to the GINA guidelines (people with mild asthma should not receive SABA alone)?

Thank you. This publication post dates our submission to Plos. We shall, of course, include such an important new reference. As indicated, our data on the adverse effects of SABA support the recommendations on p. 57 that SABA-only be no longer recommended at any stage p.17.

Preterm birth appeared more likely where only SABA were prescribed, but fell short of statistical significance, supporting recent guidelines no longer recommending ‘SABA only’ regimens for any asthma (GINA 2020 p.57).

Global Initiative for Asthma. Global Strategy for Asthma Management and Prevention. 2020, Available: https://ginasthma.org/wp-content/uploads/2020/04/GINA-2020-full-report_-final-_wms.pdf

The text on page 14, line 166-168 gives the numbers of women receiving ICS, LABA and LKA, and OCS – the proportions should also be added here.

We apologise, done

Were the women who discontinued medication the 3589 (28.3%) who received their last prescription before pregnancy (line 165)? It was not clear to me. Or was there another group who discontinued medications part way through pregnancy?

Thank you, the former is correct. We have clarified. 

and were deemed to have ‘discontinued’

The authors report that discontinuation of asthma medication before pregnancy was associated with preterm birth, but not with SGA. What is the proposed mechanism for these differing effects?

We think preterm birth is triggered by hypoxia, whereas SGA may be vasoconstriction, which is only partly determined by hypoxia. We discuss, p. 18 & 19

These [SGA] data might support a dose-response vasoconstriction hypothesis, only partly dependent on hypoxia: infants exposed to more than one medicine, likely associated with more severe asthma, were rather more vulnerable to SGA. 

The strongest predictors of premature birth were medicines discontinuation and OCS, supporting the hypothesis of a hypoxic trigger. The apparent protection against prematurity afforded by ICS suggests that ICS may have been controlling inflammation and asthma, thereby preventing hypoxia, and consequent CRH release, despite the potential to mimic endogenous corticosteroids. 

Breastfeeding prevalence at 6-8 weeks was lower in women with asthma medication before, during pregnancy, or discontinued in pregnancy. No further details were supplied in the text. This could be added, as table 6 only indicates a significant effect of asthma and unmedicated asthma in the adjusted odds ratios.

Thank you. we have added:

Prescriptions did not affect breastfeeding rates.

As the findings of this study focus on discontinuation of asthma medication, much more background information is needed on this issue, and more specifics on this are required in the methods/ results section.

Thank you. We have added information on prevalence of discontinuation, above. Discontinuation is defined in Table 1. Its impact is reported in tables 2-6, and discussed in the text. 

Some women voluntarily suspend medication during pregnancy, opting to tolerate symptoms rather than risk harming their unborn children. [58] Any uncontrolled asthma may cause chronic or intermittent hypoxia and inflammation, which adversely affect both mother and foetus, [56, 57] and may impair foetal development, causing intrauterine growth restriction, foetal distress, pre-term birth and, occasionally, death: stillbirths were more prevalent amongst women discontinuing asthma prescriptions [0.70%] when compared with UK rates 2000-2010 [0.51-0.58%] [59], and women never prescribed asthma medicines [0.38%], [S1 file, Table A1]. 

Non-adherence to asthma therapies exacerbates symptoms.[5,60] Absence of prescriptions for 9 months indicates withdrawal of therapy, as it is unlikely that stockpiled inhalers would be sufficient over this length of time. We do not know why women discontinued prescriptions or whether they experienced respiratory symptoms or they or their healthcare professionals attributed dyspnoea to pregnancy [61] or they remained well themselves, while their foetuses suffered episodes of hypoxia. Hypoxia stimulates release of placental corticotrophin-releasing hormone [CRH] and inflammatory cytokines. Placental CRH increases maternal and foetal cortisol production, which augments placental CRH release via positive feedback mechanisms, amplifying signals that may culminate in premature birth in ~5% infants. [52, 62] The strongest predictors of premature birth were medicines discontinuation and OCS, supporting the hypothesis of a hypoxic trigger

The discussion lacks critical review of the existing literature and is difficult to follow at times.

Thank you. We have revised and rechecked the literature. There are few papers reporting on discontinuation of asthma medicines in pregnancy, which cannot be conflated with non-adherence (see limitations). We should be please to follow specific advice.

The section on breastfeeding (line 303), describes these results as “neutral”; however, this was not the case as outlined in the results.

 Thank you, we have rephrased: 

Assurances [5,28] on the safety of SABA, ICS, most LABA, and OCS at <40mg per day may have contributed to the absence of a negative effect of prescriptions on breastfeeding.

 

Medicines prescribed for asthma, discontinuation and perinatal outcomes, including breastfeeding: a population cohort analysis

Short title: 

Asthma medicines, discontinuation and perinatal outcomes, including breastfeeding

Rebuttal letter PONE-D-19-17105 - [EMID:d126f82151db09f4]

30.01.20

Dear Professor Hashimoto,

Thank you for the return of our manuscript. We are very grateful for the reviewers’ support and suggestions. We have done our best to comply, as tabulated below. 

The sister paper, prepared from the same data set [1] has been published by Plos One, and has received some media coverage (below). The setting and flow diagram are inevitably the same for the two papers. This might present copyright issues if submitting to other journals. Accordingly, we are resubmitting to Plos One.

To our knowledge, we are the first to address the crucial question of medication discontinuation in pregnancy, and its impact on preterm birth, small for gestational age (SGA), and breastfeeding at 6-8 weeks. 

We feel that clinicians need to be aware of the importance of targeting women of childbearing age for medication review and additional support to reduce the risks of adverse outcomes, including exclusive formula feeding. In particular, we highlight that discontinuation of asthma medication during pregnancy increases risks of preterm delivery. Publication in PlosOne would stimulate prescribers to take a proactive approach to medication review in preconception care, an important, but neglected, component of the WHO 3rd Global Patient Safety Challenge on Medication Safety to reduce medicines-related harm by 50% by 2022 [2].

We hope that readers will concur with reviewer 2: As the asthma is a disease of many different phenotypes, severity and treatments, there is still need for more investigation such as this article.

The work has not been submitted elsewhere. 

Thank you for your time spent on this paper. 

Yours,

Sue Jordan, on behalf of all authors

[1] Jordan S, Davies GI, Thayer DS, Tucker D, Humphreys I (2019) Antidepressant prescriptions, discontinuation, depression and perinatal outcomes, including breastfeeding: A population cohort analysis. PLOS ONE 14(11): e0225133. https://doi.org/10.1371/journal.pone.0225133

Targeted News Service highlights a new study led by Swansea University’s Professor Sue Jordan on the impact of depression on pregnant women and new mothers (no link). Featured by EurekAlert. Also reported by News-Medical.net (Australia), 7thSpace (US), Doctor’s Guide (Canada).

[2] WHO 2017 Medication without harm. WHO, Geneva http://apps.who.int/iris/bitstream/10665/255263/1/WHO-HIS-SDS-2017.6-eng.pdf?ua=1&ua=1

Changes Made

Reviewer #1: 

The study covers a useful research question and the data source is appropriate for answering the key questions. From my perspective, however, the study would require substantial changes to be useful for the intended audience. Thank you.

These are my key observations: 

1) The objectives seem to be a mix of showing non causal associations and showing causal relationships. I think identifying associations is more appropriate without using more rigorous methods to account for confounding (e.g. propensity score methods). From this perspective, the current regression models would be what some people call “explanatory” models – i.e. to show various relationships – which is still useful. Some sections do, appropriately, focus on associations. Thank you, we agree that causality cannot be determined from retrospective cohorts. We have clarified our aims and methods:

To identify women likely to benefit from additional care, we... p.4

There was no randomisation, and, therefore, we explored association, rather than causality. P.4

The methods, though, describe control for confounding which would imply the goal is causal inference – i.e. to show the extent to which these medications cause the adverse pregnancy outcomes described. I would expect that all of the exclusion criteria would tend to limit generalizability, but I don’t see that they address the key confounding issues. I also don’t see that the regression models get at the key problems. When, for example, you want to know the causal association between SABA use and preterm birth, the article raises the issue of controlling for “confounding by indication.” In this case, SABAs could be used in patients with more severe asthma or with poorly controlled asthma, and therefore it is difficult to distinguish adverse effects of the drug from those of the underlying disease. 

To address confounding by indication you would need information on asthma severity (or other underlying causes of adverse outcomes) and use appropriate methods to address. The better alternative, from my perspective, is to simply treat this as descriptive data and not attempt to address confounding. You can still conduct multivariable modeling in the framework of a descriptive or explanatory regression model. You can also develop a “predictive” model but this would require a different strategy. Thank you. We now offer a definition of confounding from a standard medical text, based on an earlier Lancet series. Confounding is a blurring of effects when a variable is associated with the exposure and affects the outcome (Schulz & Grimes 2006 p.34). p.7

Restricting the dataset by excluding pregnancies exposed to AEDs, insulin, coumarins, heavy drinking or substance misuse and multiple pregnancies increases internal validity at the cost of generalisation to these groups (Schulz & Grimes 2006). P.17

Information on severity of asthma was inferred from prescriptions. As referenced, in the UK, prescriptions follow the BTS and BNF guidelines (references [5, 29]). P.5 and Table 1. Prescribers adhere to these guidelines. For example, oral corticosteroids are only prescribed for severe asthma. 

2) I would have expected the study to be conducted among women with asthma. Including all pregnant women made the interpretation more difficult. This is a population-based cohort. As healthcare databases become more available, such designs are increasingly common (Sterne et al 2016). We have added a sentence relating to their advantages p.4:

A population-based cohort was built from prospectively collected routine NHS data and analysed retrospectively. Such cohorts have minimal selection and attrition bias and no recall bias (Thygesen and Ersboll 2014). We explored association, rather than causality, as there was no randomisation.

3) I found several sections of the manuscript difficult to follow. In some cases, it was because multiple ideas were combined in the same sentence. Here is an example: “Breastfeeding [any, even if supplemented by formula feeding] at birth and 6- 8 weeks is routinely recorded by health visitors, and this information is transferred to NCCHD; data collection is more complete in some Health Board regions than others. [33] 

Breastfeeding at birth is considered an indirect measure, which may represent intention rather than practice, and is downgraded as evidence of successful breastfeeding. [34]” This section combined information on how data is collected (which I think fits better in the database description) and information on interpretation (does BF at birth correlate with later BF?), but doesn’t actually state how BF was defined for the analysis. I had trouble understanding what “indirect measure” meant (Indirect measure of longer term BF?) and what “is downgraded” meant (downgraded by other researchers?). And after all that, I still wasn’t clear on how BF was defined for the analysis, which was the main point of this section. 

As another example, the phrase “% without unknown outcome” was confusing. We have clarified, and added a reference p.6. Breastfeeding in this database is defined as ‘any breastfeeding’ (exclusive, or combined, predominantly or partially, with other feeds, including formula milk) as recorded by health visitors and entered into NCCHD (NHS Wales 2017). 

We have reworded to clarify, thank you: Breastfeeding at birth may represent intention to breastfeed, rather than actuality: a high proportion of dyads breastfeeding at birth have discontinued a week later. Therefore, this measure is downgraded as evidence of successful breastfeeding. [34] p.7

% without ‘unknown outcome’ is analogous to ‘missing or outcome unknown’. This entire table has been removed, as it is now available as supplementary material in the paper using the same dataset, but exploring antidepressants (Jordan et al 2019).

4) Starting the results and tables (e.g. Table 2) with an assessment of women excluded from your analysis was confusing. (Why focus the reader on these results if these women were not even eligible for the analysis?) I could see this assessment as a sensitivity analysis after presenting main results. This table is available in supplementary material in association with the co-submitted paper from the same data set. Therefore, it will not be submitted with this paper.

We believe this table contains very important information on the risks associated with other prescriptions, and will be of interest to those planning research in the area.

5) The results were not calculated and reported in a standard and useful way. If I look at the first 2 rows of Table 2, for example, a more useful way to present the results is:

a. Among those exposed to insulin, 213/688 (35.6%) had preterm delivery. (688 = 231+457)

b. Among those not exposed, 7863/112,628 (7.5%) had preterm delivery

c. The risk ratio (or relative risk) is then 4.48

d. The odds ratio (6.73) is technically correct but it is more difficult to interpret and could be easily interpreted as overstating the strength of the relationship. The regression models can match by using a model that reports a risk ratio. Modified Poisson is one that is easy to conduct: https://www.ncbi.nlm.nih.gov/pubmed/15033648 Log binomial can work as well although sometimes has convergence problems. This table has been removed from the paper. Readers are referred to its publication elsewhere.

We agree that odds and risk ratios can be used to calculate these differences, and readers are provided with sufficient information to make their own calculations. The numbers of subjects in table 2 are, in our opinions, too low for multivariate analyses. 

The tables report both the OR and the inverse of the OR which is confusing. In this case, the OR of 6.73 is the useful one for showing the relationship with increased risk of prematurity. The second OR (0.15) reversing the referent group is not useful. The tables should keep the referent group constant and identify the referent group (e.g. label cell as “ref”) and/or show an OR (or RR) of 1.0 for the referent group. The tables should also clearly delineate each association assessed. For example, the first 2 rows are one comparison (association between insulin and birth <37 weeks) and it would be helpful to distinguish that from subsequent comparisons. This table has been removed from the paper. Readers are referred to its publication elsewhere.

The additional data indicate to readers that data are completely reported. 

6) For each association, the comparison group should be stated clearly. In the abstract, for example, when you state that discontinuation is associated with preterm delivery, what is this in comparison to? (to those who continued all their treatments? To all other women including those without asthma or those who weren’t treated at all?) We are sorry if this is unclear. We state, p.8: For each analysis we compared the exposed with the unexposed population.

We have now added to the notes to table 2 and to the abstract: Odds ratios for adverse pregnancy outcomes were calculated for the exposed versus the unexposed population.

7) The data set seems quite old. Treatment guidelines and treatment options (such as biologics) have changed since this time. More recent data would be more relevant, if available. We agree. Were funding available, we would extend the study. However, this is the first report of the predictors of infant feeding and one of the most detailed reports available on medicine discontinuation in the ambulant population. We have used our data to identify how care can be improved, and hope that readers will consider this an important practice point. 

We have added to the ‘limitations’ p.18: Our data predate the primary care prescription of monoclonal antibodies for severe eosinophilic asthma persisting despite oral corticosteroid therapy. Manufacturers advise avoiding these medicines in pregnancy [29]; therefore, population databases may be unable to describe their safety in pregnancy for some time. 

Reviewer #2: 

The authors have found an association of adverse perinatal outcomes in primary care setting, in Brittan, Wales. It is an area with the greatest prevalence of asthma in Europe. As asthma is the most common chronic disease among pregnant women, we should be very careful on drawing conclusions. As the asthma is a disease of many different phenotypes, severity and treatments, there is still need for more investigation such as this article. Thank you. We agree, we cannot draw conclusions regarding aetiology, but we hope to suggest practice modifications. 

The authors have shown for the first time that the discontinuation of the medication of active asthma during pregnancy increases risk of preterm birth. There have been articles previously showing that unmedicated asthma has similar effects. The oral glucocorticoids are not analyzed in many of the articles evaluating asthma. The risks in the different articles are not seen similar in relation to oral glucocorticoids. The authors have shown for the first time that maternal asthma carries a risk of lower breast feeding rate for the first time. Thank you.

The authors present the literature in relation to asthma very well and their findings are in accordance to the previous literature. Thank you.

1) The terms hypoxia and asphyxia and their use should be controlled. As asthma may cause maternal hypoxia, the authors should be clear whether they mean maternal, placental, fetal, perinatal or neonatal hypoxia or asphyxia. This should be clear for the reader all times. Thank you, we have corrected. We have retained ‘asphyxia’ just once, on p.23, line 510, where complications of SGA are quoted: … range of conditions from neonatal asphyxia to mental health conditions in adulthood

2) The oral glucocorticoids are used for autoimmune diseases such as rheumatoid arthritis or inflammatory bowel diseases. These are associated with problems in pregnancies. Often people also have autoimmune tendency and suffer from several autoimmune diseases (asthma + RA+ IBD). It is also common for some patients to use short acting betasympathomimetics in pulmonary symptoms other than asthma. The authors have found association of worse prognosis associated with the oral glucocorticoid use. The authors have tried to make sure the use of oral corticosteroids is for asthma by necessitating the use of some other asthma medication. Have the authors information about the prevalence of the most common autoimmune diseases? If not, the authors should somehow comment the risk of confounding. We agree, this is an important point. Only 5 women with type 1 diabetes were prescribed oral corticosteroids (S1 file, table A1). Only 519 pregnancies were exposed to oral corticosteroids. Although we excluded the commonest co-morbidities, where pregnancy risks are known to be higher than for asthma, we acknowledge that other, rarer, co-morbidity was not excluded, and have added to our ‘limitations’ section: 

Rarer co-morbidities were not excluded. Co-morbid rheumatoid arthritis and inflammatory bowel disease were more likely amongst women prescribed oral corticosteroids, who would have received specialist care [5}. 

We agree that beta2 adrenoceptor agonists are indicated for all conditions associated with reversible airways obstruction. In this age-group, asthma is the most common. We have added to notes to Table 1:

SABA and LABA are indicated for all conditions associated with reversible airways obstruction [29]. 

And to our discussion, final paragraph:

Prescription patterns, regardless of diagnoses, particularly discontinuation in pregnancy, offer primary care nurses, midwives and nurse specialists a convenient marker to identify pregnancies at increased risk, analogous to an ‘intention to treat’ analysis: pragmatic but of unproven biological certainty. 

4) Authors state in introduction without a reference that asthma medications cross placenta and enter breastmilk. Is this true for all the asthma medications and could the sentence just be left out? As the article is not about congenital anomalies, is it necessary to add articles about them in relation to asthma? We apologise, and have added references. We suggest this short sentence supports the biological plausibility of our findings.

We have included the reference to congenital anomalies to alert future systematic reviewers, that this data set has been used for this outcome. It is important that future reviewers avoid all possible double-counting of events. 

5) Did the authors have information on just prescription or also on purchases and dispension? Please state clearly. We apologise if this is not clear. We have clarified:

We have no data on dispensing or hospital or private prescribing or internet purchasing. It is unlikely that anyone hospitalized with asthma or consulting privately would not be receiving prescriptions from primary care practitioners. Abolition of prescription charges in Wales in 2007 reduces the likelihood of private prescribing and purchasing.

6) This setting is primary care setting. The amounts of mediccations used differ a bit from the other studies. The secondary and tertiary care prescribtions on asthma medications are not included? Does this explain absence of some medication groups? Most of the asthma is managed in primary care, which make primary care settings more useful. We hope the sentence above clarifies. 

We agree, that our important findings relate to primary care, and the ‘less ill’ women. We think that even a 1% increase in preterm birth rate in these women is important, and might be avoided.

Jordan S, Davies GI, Thayer DS, Tucker D, Humphreys I (2019) Antidepressant prescriptions, discontinuation, depression and perinatal outcomes, including breastfeeding: A population cohort analysis. PLOS ONE 14(11): e0225133. https://doi.org/10.1371/journal.pone.0225133

Schulz KF and Grimes DA 2006 The Lancet Handbook of Essential concepts in Clinical Research. Elsevier, Edinburgh.

Sterne Jonathan AC, Hernán Miguel A, Reeves Barnaby C, Savović Jelena, Berkman Nancy D, Viswanathan Meera et al. ROBINS-I: a tool for assessing risk of bias in non-randomised studies of interventions BMJ 2016; 355 :i4919

NHS Wales Data Dictionary 2017 http://www.datadictionary.wales.nhs.uk/#!WordDocuments/infantfeedingdatarequirements.htm

Thygesen, L.C. & Ersbøll, A.K. When the entire population is the sample: strengths and limitations in register-based epidemiology Eur J Epidemiol (2014) 29: 551. https://doi.org/10.1007/s10654-013-9873-0

notes

Search: asthma pregnancy brestfeeding Filters: Review, Systematic Reviews, English Sort by: Most Recent

---

## [Decision Letter · Decision Letter 2]

4 Nov 2020

Medicines prescribed for asthma, discontinuation and perinatal outcomes, including breastfeeding: a population cohort analysis

PONE-D-20-02812R2

Dear Dr. Jordan,

We’re pleased to inform you that your manuscript has been judged scientifically suitable for publication and will be formally accepted for publication once it meets all outstanding technical requirements.

Kind regards,

Calistus Wilunda, DrPH

Academic Editor

PLOS ONE

Additional Editor Comments (optional):

Reviewers' comments:

Reviewer's Responses to Questions

**Comments to the Author**

1. If the authors have adequately addressed your comments raised in a previous round of review and you feel that this manuscript is now acceptable for publication, you may indicate that here to bypass the “Comments to the Author” section, enter your conflict of interest statement in the “Confidential to Editor” section, and submit your "Accept" recommendation.

Reviewer #1: All comments have been addressed

Reviewer #2: All comments have been addressed

Reviewer #3: All comments have been addressed

2. Is the manuscript technically sound, and do the data support the conclusions?

Reviewer #1: Yes

Reviewer #2: Yes

Reviewer #3: Yes

3. Has the statistical analysis been performed appropriately and rigorously? 

Reviewer #1: Yes

Reviewer #2: Yes

Reviewer #3: Yes

4. Have the authors made all data underlying the findings in their manuscript fully available?

Reviewer #1: Yes

Reviewer #2: Yes

Reviewer #3: No

5. Is the manuscript presented in an intelligible fashion and written in standard English?

Reviewer #1: Yes

Reviewer #2: Yes

Reviewer #3: Yes

6. Review Comments to the Author

Reviewer #1: The authors have responded adequately to my comments. I still do not agree with their decition to exclude stillbirth and malformations, but as long as they adequately justify their choice, then it is up to readers to decide whether they agree or not.

Reviewer #2: I believe the authors have addressed the comments and the manuscript is much improved. The aim and overall message of the manuscript is more streamlined and clearer. I would only to suggest to ensure the distinction between the group of women not on medication at all for asthma and those who have been prescribed medication for asthma but have discontinued in pregnancy is made very explicit as this is key to the overall conclusions from the results.

Reviewer #3: All of the comments raised have been fully addressed, including clarifications and stressing out some of the study`s limitations, and I believe the manuscript is not suitable for publication.

7. PLOS authors have the option to publish the peer review history of their article (what does this mean?). If published, this will include your full peer review and any attached files.

Reviewer #1: No

Reviewer #2: No

Reviewer #3: No

---

## [Editor Report · Acceptance letter]

13 Nov 2020

PONE-D-20-02812R2 

Medicines prescribed for asthma, discontinuation and perinatal outcomes, including breastfeeding: a population cohort analysis 

Dear Dr. Jordan:

I'm pleased to inform you that your manuscript has been deemed suitable for publication in PLOS ONE. Congratulations! Your manuscript is now with our production department. 

Kind regards, 

on behalf of

Dr. Calistus Wilunda 

Academic Editor

PLOS ONE